

# An ensemble of 48 physically perturbed model estimates of the 1/8° terrestrial water budget over the conterminous United States, 1980–2015

Hui Zheng[1], Wenli Fei[1,2], Zong-Liang Yang[3], Jiangfeng Wei[3,4], Long Zhao[3,5], Lingcheng Li[3,6]

[1]Key Laboratory of Regional Climate-Environment Research for Temperate East Asia, Institute of Atmospheric Physics, Chinese Academy of Sciences, Beijing, 100029, China

[2]University of Chinese Academy of Sciences, Beijing, 100049, China

[3]Department of Geological Sciences, John A. and Katherine G. Jackson School of Geosciences, the University of Texas at Austin, Austin, Texas, 78705, USA

[4]Collaborative Innovation Center on Forecast and Evaluation of Meteorological Disasters/Key Laboratory of Meteorological Disaster, Ministry of Education/International Joint Research Laboratory on Climate and Environment Change, Nanjing University of Information Science and Technology, Nanjing, 210044, China

[5]School of Geographical Sciences, Southwest University, Chongqing, 400715, China

[6]Pacific Northwest National Laboratory, Richland, Washington, 99354, USA

*Correspondence to*: Zong-Liang Yang (liang@jsg.utexas.edu)

**Abstract.** Terrestrial water budget (TWB) data over large domains are of high interest for various hydrological applications. Spatiotemporally continuous and physically consistent estimations of TWB rely on land surface models (LSMs). As an augmentation of the operational North American Land Data Assimilation System Phase 2 (NLDAS-2) four-LSM ensemble, this study presents a 48-member perturbed-physics ensemble configured from the Noah LSM with multi-physics options (Noah-MP). The 48 Noah-MP physics configurations are selected to give a representative cross-section of commonly used LSMs for parameterizing runoff, atmospheric surface layer turbulence, soil moisture limitation on photosynthesis, and stomatal conductance.

The ensemble simulated the 1980–2015 monthly TWB over the conterminous United States (CONUS) at a 1/8° spatial resolution. Simulation outputs include total evapotranspiration and its constituents (canopy evaporation, soil evaporation, and transpiration), runoff (the surface and subsurface components), as well as terrestrial water storage (snow water equivalent, four-layer soil water content from the surface down to 2 m, and the groundwater storage anomaly). This dataset is available at https://doi.org/10.5281/zenodo.7109816 (Zheng et al., 2022). Evaluations carried out in this study and previous investigations show that the ensemble performs well in reproducing the observed terrestrial water storage, snow water equivalent, soil

moisture, and runoff. Noah-MP complements the NLDAS models well, and adding Noah-MP consistently improves the NLDAS estimations of the above variables in most areas of CONUS. Besides, the perturbed-physics ensemble facilities the identification of model deficiencies. The parameterizations of shallow snow, lakes, and near-surface atmospheric turbulence should be improved in future model versions.

## 1.    Introduction

Estimates of the terrestrial water budgets (TWBs)—evapotranspiration, runoff, terrestrial water storage, and their constitutes—over continental domains are of high interest for a broad range of hydrological applications. Publicly available data have been applied to investigate the state of the terrestrial water cycle (Trenberth and Fasullo, 2013a; Rodell et al., 2015; Scanlon et al.,

2018; Yin and Roderick, 2020; Pascolini-Campbell et al., 2021); to understand the interactions among hydrological processes, vegetation, climate, and human activities (Trenberth and Fasullo, 2013b; LaFontaine et al., 2015; Ward et al., 2014; Levia et al., 2020); to examine the availability and variability of water resources and use (Wu et al., 2021; Hejazi et al., 2014; Scanlon et al., 2012; Voss et al., 2013; Lv et al., 2019; Le et al., 2011; Rodell et al., 2009); and to assess the risk of extreme events such as droughts (Peters-Lidard et al., 2021; Prudhomme et al., 2014; Dai, 2013; Su et al., 2021) and floods (Emerton et al., 2017;

Lin et al., 2018).

As the applications have expanded, the availability of TWB estimates has increased rapidly (Peters-Lidard et al., 2018; Saxe et al., 2021; Zhang et al., 2018). Commonly used estimation methods include remote sensing, in-situ observations, and model simulations (Saxe et al., 2021; McCabe et al., 2017; Pan et al., 2012; Gao et al., 2010; Trenberth et al., 2007). Among these

methods, land surface models (LSMs) are apt for continuously producing physically consistent TWBs over a large domain and long period, and their characteristics are particularly favorable for certain circumstances. For instance, LSMs can estimate various TWB components simultaneously; whereas, for some components, such as runoff (Lin et al., 2019; Beck et al., 2017), root-zone soil moisture (Xia et al., 2015a, b), and transpiration (Lian et al., 2018), direct remote sensing is either unavailable or highly uncertain. Additionally, LSMs are valuable in remote or topographically complex regions because of the sparseness

of in-situ observations (Kim et al., 2021). Estimations based on remote sensing and in-situ observations are often impeded by scale mismatches and observation gaps, whereas these issues are rarely an impairment for LSM simulations. Besides, LSM simulations can complement remote sensing and in-situ observations well. Combinations of estimates from different techniques can improve the estimation accuracy (Zhang et al., 2018; Pan et al., 2012; Zhao and Yang, 2018), while comparisons between model-simulated estimates and observations can reveal the impacts of human activities (Zaussinger et al., 2019) and

underground processes (Zheng et al., 2020).





Several operational LSM simulation systems have been set up over different regions of the globe (Xia et al., 2019; Shi et al., 2011; Carrera et al., 2015; Rodell et al., 2004). The systems combine an ensemble of LSMs to utilize the competitive strengths of different LSMs and to eliminate the weakness associated with individual ones. Among them, the North American Land Data Assimilation System (NLDAS) (Xia et al., 2012b, a; Mitchell et al., 2004) stands as a pioneering and successful one. The NLDAS phase 2 (NLDAS-2) operates over the conterminous United States (CONUS) from 1979 to near real-time at a spatial resolution of 1/8°. The system generates a set of multisource synthesized data of surface meteorology, vegetation, and soils, and uses them to drive an ensemble of four different LSMs. The four LSMs—namely Noah version 2.8 (Ek et al., 2003; Chen and Dudhia, 2001a, b; Chen et al., 1997), Variable Infiltration Capacity (VIC) version 4.0.3 (Liang et al., 1994), Mosaic (Koster and Suarez, 1992), and Sacramento Soil Moisture Accounting (SAC) model (Burnash et al., 1973), were selected to give a good cross-section of the diverse range of LSMs with their different physical parameterizations (Mitchell et al., 2004). The models have varying strengths and weaknesses in process parameterizations and modeling skills (Kumar et al., 2017). An ensemble of multiple models can produce an aggregated estimate that outperforms most of the individual ensemble constituents (Fei et al., 2021; Beck et al., 2017; Guo et al., 2007; Ajami et al., 2007) and quantify the estimation uncertainty resulting from different model formulations (Troin et al., 2021; Cloke and Pappenberger, 2009). Evaluations of the NLDAS-2 four-LSM ensemble estimates have shown satisfactory performance in matching the observed evapotranspiration (ET) (Zhang et al., 2020; Xia et al., 2012b; Kumar et al., 2018), runoff (Xia et al., 2012a), and soil moisture (Xia et al., 2015a, b).

This study enriches the NLDAS-2 four-model ensemble with 48 perturbed-physics configurations of the Noah LSM with multi-physics options (Noah-MP) (Niu et al., 2011; Yang et al., 2011). Noah-MP has more physically realistic representations of the vertical stratification than the NLDAS-2 models have. A column of land in Noah-MP consists of a vegetation canopy layer, three snowpack layers, four soil layers, and a groundwater component (Niu et al., 2011). Conceptual (e.g., the five water tanks of SAC) and lumped (e.g., the combined vegetation-soil surface layer of Noah) representations of the stratification of vegetation and soil, as used in the NLDAS-2 models, are minimized. Moreover, Noah-MP has a more comprehensive representation of various land surface processes that are evident at different depths. The modeled processes include snow accumulation and ablation, infiltration, percolation, retention, freeze–thaw of snow or soil water, groundwater recharge/discharge, and energy constraints (Niu et al., 2011). These improvements in vertical stratification and process parameterizations are expected to better estimate TWBs. Indeed, previous comparisons between Noah-MP and the four NLDAS-2 LSMs have shown that Noah-MP is comparable or better when it comes to estimating soil moisture (Cai et al., 2014a), runoff (Cai et al., 2014a; Fei et al., 2021), and ET (Zhang et al., 2020). Such results have encouraged computationally expensive runs of Noah-MP, as performed in this study.

The enrichment of this study also features a single-model perturbed-physics ensemble, which is different from the widely used multi-model ensemble approach. The Noah-MP ensemble is constructed by shuffling the available parameterization options

of several selected processes. The ensemble size grows exponentially as a multiplication of the available parameterization options of different processes (Yang et al., 2011; Zhang et al., 2016; Gan et al., 2019). A large ensemble should give a broad cross-section of feasible model formulations to account for the model uncertainty in TWB estimation (Telteu et al., 2021; Mitchell et al., 2004) and is critical for a statistically reliable estimation of the probability of hydrological events such as floods and droughts (Troin et al., 2021). The single-model perturbed-physics ensemble also facilitates uncertainty attribution and
reduction. The ensemble consists of pairs that are different in the parameterization of one process and the same for another. The impacts of the process's parameterizations can be pinpointed from the model predictive variations. Variance analysis can be applied to quantify the contribution of the parameterization of the process and compare the relative importance of two processes (Zheng et al., 2019; Clark et al., 2011), and such a quantification could inform further model development to reduce the model uncertainty. However, there are pitfalls unique to the single-model perturbed-physics ensemble.


In this study, we assess the spread among the ensemble members, reveal the difference with the NLDAS models, and evaluate the performance against various observations. The paper is organized as follows. Section 2 presents the information necessary for using the dataset, including the dataset variables, file organization, and the source data and models used for data generation. Section 3 describes the intercomparison and evaluation methods, along with the reference datasets. Section 4 presents the
results and related discussion. Finally, after stating the data availability in Section 5, Section 0 concludes this study.

## 2.    Data description

The dataset contains gridded water budget variables over CONUS. Section 2.1 describes the dataset variables and their physical relationships. The 48 Noah-MP physics configurations used to create the dataset are detailed in Section 2.2. Section 2.3 brief the atmospheric forcing, static parameters of vegetation and soil, and simulation settings.

### 2.1.    Dataset variables

**Table 1** lists the dataset variables. The variables are available at each 1/8° grid points in NLDAS-2, indicated by a land–water mask ($X$). The surface water budgets of each grid cell are represented as follows:

Neglecting horizontal water exchange between adjacent grids, the water budget closure can be obtained among the precipitation
($P$, kg m$^{-2}$ s$^{-1}$), ET ($E$), runoff ($R$), and terrestrial water storage change ($\Delta W$) (Zheng et al., 2020):

$$P = E + R + \Delta W, \tag{1}$$

where precipitation ($P$) is from NLDAS-2 (described in Section 1) and used as the model input.



Noah-MP resolves the components of the water budget terms of equation (1). ET ($E$) consists of canopy evaporation ($E_{can}$),

ground evaporation ($E_{gnd}$), and transpiration ($E_{tran}$):

$$E = E_{can} + E_{gnd} + E_{tran}. \tag{2}$$

Runoff ($R$) has a surface ($R_{srf}$) and subsurface ($R_{sub}$) component:

$$R = R_{srf} + R_{sub}. \tag{3}$$

Terrestrial water storage (TWS, $W$) is the sum of snow water equivalent (SWE, $W_{snow}$), groundwater storage in unconfined

aquifers ($W_{gw}$), and soil water content in the four model layers ($W_{soil,i}$, kg m$^{-2}$):

$$W = W_{snow} + W_{gw} + \sum_{i=1}^{N_{soil}} W_{soil,i}, \tag{4}$$

where $N_{soil} = 4$ is the number of soil layers. Soil water storage ($W_{soil,i}$) is not included in the dataset but can be calculated

from the volumetric water content ($w_i$) as follows:

$$W_{soil,i} = \rho_{wat} \cdot w_i \cdot \Delta z_i \ for \ i = 1, \dots, 4, \tag{5}$$

where $\rho_{wat} = 1000$ kg m$^{-3}$ is the water density; and $\Delta z_{soil,1} = 0.1$ m, $\Delta z_{soil,2} = 0.3$ m, $\Delta z_{soil,3} = 0.6$ m, and $\Delta z_{soil,4} = 1$ m

are the thicknesses of the four soil layers.

## 2.2.   The 48 Noah-MP physics configurations

The Noah-MP LSM version 3.6 is used. The 48 physics configurations ($48 = 4 \times 2 \times 3 \times 2$) are generated by combining four

runoff parameterizations (Sections 2.2.1-2.2.4), two parameterizations of stomatal conductance (Sections 2.2.5 and 2.2.6),

three parameterizations of soil moisture stress factor (Section 2.2.7), and two parameterizations of near-surface atmospheric

turbulence (Sections 2.2.8 and 2.2.9). In addition to the explanations of the adopted parameterizations and their acronyms in

Zheng et al. (2019, Table 1), the following subsections detail the formulations of the parameterization schemes.

### 2.2.1.   SIMGM runoff parameterization scheme

SIMGM is a TOPMODEL-based runoff parameterization scheme (Niu et al., 2007). This scheme parameterizes runoff ($R_{srf}$

and $R_{sub}$) as an exponential function of groundwater table depth ($z_{wt}$, m, positive down) as follows.

$$R_{srf} = Q_{soil,srf}(1 - f_{frz,1})f_{sat} + f_{frz,1}, \tag{6}$$

$$f_{sat} = f_{sat,max} \exp[-0.5f(z_{wt} - z_{bot})], \tag{7}$$

$$R_{sub} = \left[1 - \max_{i=1,\dots,4}(f_{frz,i})\right] R_{sub,max} \exp[-\Lambda - f(z_{wt} - z_{bot})], \tag{8}$$

where $Q_{soil,srf}$ is the water incident on the soil surface (the sum of precipitation throughfall, snowmelt, and dewfall; kg m$^{-2}$ s$^{-1}$);

$f_{frz,i}$ is the fractional frozen area of the $i$th soil layer (m$^2$ m$^{-2}$), which is parameterized using the frozen water content of the

soil layer following Niu & Yang (2006); $f_{sat}$ is the saturation fraction of the grid cell (m$^2$ m$^{-2}$); $z_{bot}$ is the depth of the soil



column bottom (2 m in this study); and $z_{wt}$ is the groundwater table depth (m), which is converted from the groundwater storage by a specific-yield parameter. The groundwater storage is predicted using a dynamic groundwater model interacting with the soil column bottom (Niu et al., 2007).


The scheme has four calibratable parameters: (1) $f_{sat,max}$, the maximum saturated area fraction (m² m⁻²), which is defined as the cumulative distribution function of the topographic index when the grid-cell-mean water table depth is zero; (2) $f$, a runoff decay factor (unitless); (3) $R_{sub,max}$, the maximum subsurface runoff when the grid-cell-mean water table depth is zero (kg m⁻² s⁻¹); and (4) $\Lambda$, the grid-cell-mean topographic index (unitless). In this study, the parameters have the following values: $f_{sat,max} = 0.38$ m² m⁻², $f = 6$, $R_{sub,max} = 5$ kg m⁻² s⁻¹, and $\Lambda = 10.5$.

### 2.2.2. SIMTOP runoff parameterization scheme

SIMTOP is also a TOPMODEL-based runoff parameterization scheme, the same as SIMGM (equations (6)–(8)). The major difference between SIMTOP and SIMGM is that SIMTOP parameterizes the groundwater table depth ($z_{wt}$) using the soil liquid water content by assuming the water head is at equilibrium throughout the soil column down to the water table (Niu et al., 2005). Although SIMTOP and SIMGM share the same conceptual model of runoff generation, implementation differences exist. First, in contrast to equations (7) and (8), SIMTOP does not use the soil column bottom depth ($z_{bot}$) in calculating the saturation area fraction ($f_{sat}$) and subsurface runoff:

$$f_{sat} = f_{sat,max} \exp(-0.5 f z_{wt}), \tag{9}$$

$$R_{sub} = \left[1 - \max_{i=1,\dots,4}\left(f_{frz,i}\right)\right] R_{sub,max} \exp(-\Lambda - f z_{wt}). \tag{10}$$

Second, parameter values are slightly different for the runoff decay factor and maximum subsurface runoff: $f = 2$ and $R_{sub,max} = 4$ kg m⁻² s⁻¹.

### 2.2.3. NOAHR runoff parameterization scheme

NOAHR parameterizes surface runoff ($R_{srf}$) as infiltration excess:

$$R_{srf} = Q_{soil,srf} - Q_{soil,in}, \tag{11}$$

where $Q_{soil,in}$ is the infiltration into the soil (kg m⁻² s⁻¹). The infiltration is derived from the approximate solution to the Richards equation following Philip (1969) with additional considerations of the spatial variability of precipitation and infiltration capacity. By assuming exponential and independent distributions of precipitation and infiltration capacity within a model grid cell, NOAHR formulates the soil infiltration as follows:

$$Q_{soil,in} = Q_{soil,srf} \frac{I_c}{Q_{soil,srf}\Delta t + I_c}, \tag{12}$$

$$I_c = w_d[1 - \exp(-K_{\Delta t}\Delta t)], \tag{13}$$



$$w_d = \sum_{i=1}^{N_{soil}} \rho_{wat}\left(w_{max,i} - w_i\right)\Delta z_{soil,i}, \tag{14}$$

where $I_c$ is the soil infiltration capacity of the model grid cell (kg m$^{-2}$), $w_d$ is the water deficit of the soil column (kg m$^{-2}$), and $\Delta t$ is the model time step (s). $K_{\Delta t}$ is a calibratable parameter. Following Chen & Dudhia (2001a), the parameter is assumed as propositional to the saturated hydraulic conductivity of the first soil layer ($K_{sat,1}$, kg m$^{-2}$ s$^{-1}$):

$$K_{\Delta t} = \frac{K_{\Delta t,ref}}{k_{ref}}K_{sat,1}, \tag{15}$$

where $K_{\Delta t,ref}$ and $k_{ref}$ are two parameters. In Noah-MP (and Noah), $K_{\Delta t,ref} = \frac{3}{86400}$ s$^{-1}$, and $k_{ref} = 2 \times 10^{-3}$ kg m$^{-2}$ s$^{-1}$. $K_{sat,1}$ is assigned using a soil parameter lookup table according to the soil texture type.

NOAHR assumes free drainage at the soil column bottom. The subsurface runoff is calculated as

$$R_{sub} = \alpha_{slope}K_4, \tag{16}$$

where $\alpha_{slope}$ is the terrain slope index, which is arbitrarily given as $0.1$ in the adopted version of Noah-MP. $K_4$ is the hydraulic conductivity of the bottom soil layer, which is parameterized following Clapp and Hornberger (1978).

### 2.2.4. BATS runoff parameterization scheme

The BATS runoff scheme parameterizes surface runoff ($R_{srf}$) as a function of soil wetness (Yang and Dickinson, 1996):

$$R_{srf} = Q_{soil,t}\left(1 - f_{frz,1}\right)f_{sat} + f_{frz,1}, \tag{17}$$

$$f_{sat} = \theta^4, \tag{18}$$

$$\theta = \frac{\sum_{i=1}^{N_{soil}} \frac{w_i}{w_{sat,i}}\Delta z_{soil,i}}{\sum_{i=1}^{N_{soil}} \Delta z_{soil,i}}, \tag{19}$$

where $\theta$ is the averaged wetness throughout the soil column (m$^3$ m$^{-3}$).

Similar to NOAHR, the BATS scheme also assumes a free drainage boundary condition at the soil column bottom. Subsurface runoff ($R_{sub}$) is parameterized as follows:

$$R_{sub} = \left(1 - \max_{i=1,\dots,4}\left(f_{frz,i}\right)\right)K_4. \tag{20}$$



### 2.2.5.    Ball–Berry scheme of stomatal resistance

Leaf stomata are the small pores typically found on the underside of leaves. They control the gas exchange of $CO_2$, $H_2O$, and
$O_2$ between the internal leaf structure and the external atmosphere. In LSMs, the opening and closing of the stomata are characterized by stomatal conductance.

The Ball–Berry scheme for parameterizing the stomatal conductance ($g_s$) for $H_2O$ is as follows:

$$g_s = m \frac{A}{c_s} \frac{e_s}{e_i} P_{atm} + b,\tag{21}$$

where $g_s$ is the leaf stomatal conductance (µmol m$^{-2}$ s$^{-1}$), $m$ is a vegetation-type dependent parameter (unitless), $A$ is the leaf photosynthesis rate, $c_s$ is the $CO_2$ partial pressure at the leaf surface (Pa), $e_s$ is the water vapor pressure at the leaf surface (Pa), $e_i$ is the saturated water vapor at the stomata (Pa), $P_{atm}$ is the ambient air pressure (Pa), and $b$ is the stomatal conductance at zero photosynthesis (µmol m$^{-2}$ s$^{-1}$). The parameters $m$ and $b$ are assigned from a lookup table using the vegetation type.

### 2.2.6.    Jarvis scheme of stomatal resistance

The Jarvis scheme for parameterizing the canopy resistance ($R_c$) based on the product of four stress factors (s m$^{-1}$) is calculated as follows (Chen et al., 1996; Sellers et al., 1996; Jarvis, 1976):

$$R_c = R_{c,min} \frac{1}{f_1 f_2 f_3 \beta},\tag{22}$$

$$f_1 = \frac{\frac{R_{c,min}}{R_{c,max}} + f}{1 + f},\tag{23}$$

$$f = 0.55 \frac{2R_g}{R_{gl}},\tag{24}$$


$$f_2 = \frac{1}{1 + h_s[q_{sat}(T_l) - q_a]},\tag{25}$$

$$f_3 = 1 - 0.0016(T_{ref} - T_l)^2,\tag{26}$$

where $f_1$, $f_2$, and $f_3$ are the stress factors of solar radiation, vapor pressure deficit, and air temperature, respectively (unitless), which are unitless and range from 0 to 1; $\beta$ is the soil moisture stress factor, which is detailed in Section 2.2.7; $R_g$ is the incoming solar radiation (W m$^{-2}$) for unit leaf area index; $T_l$ is leaf temperature (K); $q_{sat}(T_l)$ is the saturated specific humidity at the temperature of $T_l$ (kg kg$^{-1}$); and $q_a$ is the ambient specific humidity (kg kg$^{-1}$). The scheme has five parameters: $R_{c,min}$
the minimum stomatal resistance (s m$^{-1}$) per unit leaf area index; $R_{c,max}$ the maximum resistance; $R_{gl}$ a radiation scaling factor (unitless); $h_s$ a humidity scaling factor (unitless); $T_{ref}$ the optimum temperature (K). Among these parameters, $R_{c,min}$, $R_{gl}$,



and $h_s$ are assigned using a vegetation-parameter lookup table, while $R_{c,max}$ and $T_{ref}$ are preassembly-assigned to 5000 s m$^{-1}$ and 298 K, respectively.

### 2.2.7. Three soil moisture stress factor schemes

The NOAHB scheme parameterizes the soil moisture stress factor controlling transpiration ($\beta$-factor) as a function of soil moisture, which is calculated as follows:

$$\beta = \sum_{i=1}^{N_{root}} \frac{\Delta z_{soil,i}}{z_{root}} \min\left(1, \frac{\theta_i - \theta_{wilt}}{\theta_{ref} - \theta_{wilt}}\right), \tag{27}$$

where $N_{root}$ is the total number of soil layers that contain roots, $z_{root}$ is the total depth of the root zone layer (m), and $\theta_i$ is the volumetric soil moisture of the $i$-th soil layer (m$^3$ m$^{-3}$). NOAHB has two parameters: $\theta_{sat}$, the saturated volumetric soil moisture (m$^3$ m$^{-3}$); and $\theta_{wilt}$, the wilting volumetric soil moisture (m$^3$ m$^{-3}$).

The CLM scheme (Oleson et al., 2004) parameterizes the $\beta$-factor as a function of soil matric potential, which is calculated as follows:

$$\beta = \sum_{i=1}^{N_{root}} \frac{\Delta z_{soil,i}}{z_{root}} \min\left(1, \frac{\psi_{wilt} - \psi_i}{\psi_{wilt} - \psi_{sat}}\right), \tag{28}$$

where $\psi_i$ is the water pressure head of the $i$th soil layer (m), and $\psi_i$ is converted from $\theta_i$ using the formula of Clapp and Hornberger (1978). CLM has two parameters: $\psi_{sat}$, the saturated water pressure head (m), and $\psi_{wilt}$, the wilting pressure head (m).

The SSiB scheme (Xue et al., 1991) also parameterizes the $\beta$-factor as a function of the soil pressure head, similar to CLM. However, the formula is different, as follows:

$$\beta = \sum_{i=1}^{N_{root}} \frac{\Delta z_{soil,i}}{z_{root}} \min\left[1, 1 - \exp\left(-c_2 \ln\left(\frac{\psi_{wilt}}{\psi_i}\right)\right)\right]. \tag{29}$$

SSiB has two parameters: $\psi_{wilt}$, the wilting pressure head (m); and $c_2$, a unitless coefficient.

In Noah-MP version 3.6, the parameters $\theta_{sat}$, $\theta_{wilt}$, and $\psi_{sat}$ are assigned using a soil parameter lookup table (Chen and Dudhia, 2001a, Table 2); $\psi_{wilt}$ is −150 m, independent of vegetation and soil types (Niu et al., 2011); $c_2$ is assumed constant at 5.8, whereas in SSiB, this parameter varies with vegetation type (Xue et al., 1991, Table 2).



### 2.2.8. Chen97 near-surface turbulence scheme

The Chen97 scheme (Chen et al., 1997) parameterizes the surface exchange coefficient for heat ($C_h$) as follows:

$$C_h = \kappa^2 \left[\ln\left(\frac{z}{z_{0m}}\right) - \Psi_m\left(\frac{z}{L}\right) + \Psi_m\left(\frac{z_{0m}}{L}\right)\right]^{-1} \left[\ln\left(\frac{z}{z_{0h}}\right) - \Psi_h\left(\frac{z}{L}\right) + \Psi_h\left(\frac{z_{0h}}{L}\right)\right]^{-1},$$ (30)

where $\kappa = 0.4$ is the von Kármán constant; $L$ is the Monin–Obukhov (M–O) length (m); $z$ is the reference height (m); $\Psi_m$ and $\Psi_h$ are the similarity theory–based stability functions for momentum and heat, respectively; $z_{0m}$ the roughness length for momentum (m), depends on the land cover/land-use type; and $z_{0h}$ is the roughness length for heat (m). Niu et al. (2011) parameterized $z_{0h}$ as $z_{0h} = z_{0m} \exp\left(-\kappa C \sqrt{Re^*}\right)$, where $C = 0.1$ and $Re^*$ is the roughness Reynolds number. However, in the code of Noah-MP version 3.6, $z_{0h} = z_{0m}$.

### 2.2.9. M–O near-surface turbulence scheme

The M–O scheme is based on the M–O similarity theory (Brutsaert, 1982), which parameterizes $C_h$ as follows:

$$C_h = \kappa^2 \left[\ln\left(\frac{z - d_0}{z_{0m}}\right) - \Psi_m\left(\frac{z - d_0}{L}\right)\right]^{-1} \left[\ln\left(\frac{z - d_0}{z_{0h}}\right) - \Psi_h\left(\frac{z - d_0}{L}\right)\right]^{-1},$$ (31)

where $z_{0h} = z_{0m}$, and $d_0$ is the zero-displacement height (m),

$$d_0 = 0.65 z_{ct},$$ (32)

where $z_{ct}$ is the canopy top height (m).

### 2.3. Domain, temporal span, atmospheric forcings, and static parameters

The simulation domain covers the all of CONUS (25°–53°N, 125°–67°W), which is also called the NLDAS-2 testbed (Xia et al., 2012a, b). The simulations were performed at a spatial resolution of 0.125°, which is the same as for NLDAS-2 models.

The hourly NLDAS-2 atmospheric forcings at a spatial resolution of 0.125° were used to drive the 48 Noah-MP configurations. This study used seven forcing variables: downward solar radiation, downward longwave radiation, air temperature, surface pressure, specific humidity, wind speed, and precipitation rate. The static datasets, including topography (https://ldas.gsfc.nasa.gov/nldas/elevation), predominant vegetation class (https://ldas.gsfc.nasa.gov/nldas/vegetation-class), and soil texture type (https://ldas.gsfc.nasa.gov/nldas/soils), are also the same as in NLDAS-2. We used the default Noah-MP lookup tables to convert the input vegetation and soil types to parameter values.

The simulation spans 36 years from January 1980 to December 2015 at a time step of 15 minutes. The initial states of each Noah-MP configuration were obtained from a 100-year-long spin-up over the single year of 1979. Details of the simulation settings and spin-up run can be found in Section 2.3 of Zheng et al. (2019) and Section 2.2 of Fei et al. (2021).





## 3. Intercomparison and evaluation methods

We have previously evaluated the runoff and compared it with NLDAS (Fei et al., 2021). Therefore, this study focuses on assessing TWS, SWE, soil moisture, and ET. We also examine the spread among the Noah-MP configurations. The evaluations and intercomparisons are performed for 12 River Forecast Centers (RFCs): Northeast (NE), Mid-Atlantic (MA), Ohio (OH),

Lower Mississippi (LM), Southeast (SE), North Central (NC), Northwest (NW), Arkansas (AB), Missouri (MB), West Gulf (WG), California–Nevada (CN), and Colorado (CB). Figure S1 displays the geographical delineation of the RFCs. More details on the RFCs, such as their multi-year average precipitation, potential evaporation, and topography, can be found in Fei et al. (2021, Figure 1).

The intercomparison and evaluations were conducted at three different time scales—namely the long-term climatology, annual cycle, and interannual anomaly. Section 3.1 details how the temporal variations at different time scales are derived. Ensemble spread is defined in Section 3.2, and a ranking of the spread magnitude is assigned relative to the temporal variation following the Global Soil Wetness Project (GSWP) (Dirmeyer et al., 2006). We utilized the Taylor diagram and Taylor Skill Score (TSS) to measure the performance of Noah-MP against various reference datasets. The evaluation methods are shown in Section 3.3,

and the reference datasets are described in Section 3.5. In addition to the inter-comparisons and evaluations, Section 3.4 introduces the Sobol' sensitivity index for the process sensitivity analysis.

### 3.1. Climatology, annual cycle, and interannual anomaly

For each variable $r_{y,m}$ of the dataset (month $m$ of the year $y$, $m = 1 \ldots 12$, $y = 1 \ldots Y$), the multi-year averaged climatology ($r_{clim}$), annual cycle ($r_{ancy,m}$), and interannual anomaly ($r_{anom,y,m}$) are calculated as follows:

$$r_{clim} = \frac{1}{12Y} \sum_{y=1}^{Y} \sum_{m=1}^{12} r_{y,m} \,, \tag{33}$$

$$r_{ancy,m} = \frac{1}{Y} \sum_{y=1}^{Y} r_{y,m} - r_{clim}, \tag{34}$$

$$r_{anom,y,m} = r_{y,m} - r_{ancy,m} - r_{clim}. \tag{35}$$

The temporal variability of $r_{y,m}$ ($\sigma_{total}$), $r_{ancy,m}$ ($\sigma_{ancy}$), and $r_{anom,y,m}$ ($\sigma_{anom}$) are derived as follows:

$$\sigma_{total} = \sqrt{\frac{1}{12Y} \sum_{y,m} (r_{y,m} - r_{clim})^2} \,, \tag{36}$$

$$\sigma_{ancy} = \sqrt{\frac{1}{12} \sum_{m} (r_{ancy,m} - r_{clim})^2} \,, \tag{37}$$





$$\sigma_{anom} = \sqrt{\frac{1}{12Y}\sum_{y,m}(r_{y,m} - r_{ancy,m})^2}\,. \tag{38}$$

### 3.2.   Spread among the ensemble members

The spread among $N$ ensemble members is measured by standard deviations following GSWP (Dirmeyer et al., 2006):

$$\sigma(r_t) = \sqrt{\frac{1}{N}\sum_{n=1}^{N}(r_{n,t} - \bar{r}_t)^2}\,, \tag{39}$$

$$\bar{r}_t = \frac{1}{N}\sum_{n=1}^{N}r_{n,t}\,, \tag{40}$$

where $\bar{r}_t$ refers to the ensemble mean.

Measures of the ensemble spread at different time scales are obtained as temporal averages over the time horizon as follows:

$$\sigma_{lss,total} = \frac{1}{T}\sum_{t=1}^{T}\sigma(r_t)\,, \tag{41}$$

$$\sigma_{lss,ancy} = \frac{1}{12}\sum_{m}^{12}\sigma(r_{ancy,m})\,, \tag{42}$$

$$\sigma_{lss,anom} = \frac{1}{12Y}\sum_{y=1}^{Y}\sum_{m=1}^{12}\sigma(r_{anom,y,m})\,, \tag{43}$$

where $\sigma_{lss,total}$ is the ensemble spread defined the same as in GSWP, $\sigma_{lss,ancy}$ is the ensemble spread for the modeled annual cycle, and $\sigma_{lss,anom}$ is the ensemble spread for the modeled interannual anomaly.

To compare the ensemble spread for various dataset variables at different time scales, the ratio ($R$) of the ensemble spread to the temporal variability is calculated: (1) $R = \sigma_{lss\_total}/\sigma_{total}$, for the whole time series; (2) $R = \sigma_{lss\_ancy}/\sigma_{ancy}$, for the annual cycle; and (3) $R = \sigma_{lss\_anom}/\sigma_{anom}$, for the interannual anomaly. Similar to GSWP (Dirmeyer et al., 2006), we grade the ratio as follows: "A" for $R < 0.316$; "B" for $0.316 \leq R < 1$; "C" for $1 \leq R < 3.16$; "D" for $3.16 \leq R < 10$; and "E" for $R > 10$. A lower (higher) $R$ or a higher (lower) grade denotes a lower (higher) ensemble spread.

### 3.3.   Taylor diagram and skill score

The Taylor diagram (Taylor, 2001) is a graphical representation of how closely a model simulation matches observations in terms of correlation coefficient ($R$), normalized unbiased root-mean-square error (nuRMSE), and normalized standard





deviation ($\hat{\sigma}_f$). The TSS is an index that measures the distance between a model simulation and the observations in the Taylor

diagram. The TSS is defined as follows:

$$TSS = \frac{4(1+R)}{\left(\hat{\sigma}_f + \frac{1}{\hat{\sigma}_f}\right)^2 (1+R_0)}, \tag{44}$$

$$\hat{\sigma}_f = \frac{\sigma_f}{\sigma_o}, \tag{45}$$

where $\sigma_f$ and $\sigma_o$ are the standard deviations of the model simulation and the observation, and $R_0$ is the maximum correlation

coefficient attainable (in this study, $R_0 = 1$). The value range of TSS is from 0 to 1. A higher TSS indicates a higher overall

performance of model prediction with reference to the observations.

### 3.4.   Sobol' total sensitivity analysis

The sensitivity of the Noah-MP ensemble to a physical process can be quantified by the Sobol' total sensitivity index (Saltelli

and Sobol', 1995; Zheng et al., 2019). The Sobol' total sensitivity index measures the proportion of the variance of different

processes to the total variance, which is defined as follows:


$$S_\Lambda = \frac{E_{\sim\Lambda}(Var_\Lambda(Y|\sim\Lambda))}{Var(Y)}, \tag{46}$$

where $S_\Lambda$ is the Sobol' total sensitivity index for one process $\Lambda$; $\sim\Lambda$ represents the other processes except for $\Lambda$; $Y$ represents

the 48 Noah-MP ensemble members; $Var(Y)$ is the variance of $Y$; $Var_\Lambda(Y|\sim\Lambda)$ denotes the variance among different

parameterization schemes of the process $\Lambda$, and $E_{\sim\Lambda}$ denotes the arithmetic average across all combinations of the other

processes except for $\Lambda$. Detailed calculations and examples can be found in Zheng et al. (2019).

### 3.5.   Reference data

#### 3.5.1.   Terrestrial water storage

We use the $1° \times 1°$ monthly Gravity Recovery and Climate Experiment (GRACE) land water-equivalent-thickness surface

mass anomaly, level-3, Release 6.0, version 04, as the reference of TWS ($W$ in Table 1). The GRACE products are derived

from the gravity anomaly measured by twin satellites and have had the signals from factors such as glacial isostatic adjustment

and tides removed. The GRACE products from different processing centers are slightly different. Therefore, to reduce the

noise of different products (Sakumura et al., 2014), we use the arithmetic average of the products from three centers—namely

GeoForschungsZentrum   Potsdam   (or   the   German   Research   Center   for   Geosciences)   (Landerer,   2021a)

(https://podaac.jpl.nasa.gov/dataset/TELLUS_GRAC_L3_GFZ_RL06_LND_v04); the Center for Space Research at the

University   of   Texas,   Austin   (Landerer,   2021b)

(https://podaac.jpl.nasa.gov/dataset/TELLUS_GRAC_L3_CSR_RL06_LND_v04); and NASA's Jet Propulsion Laboratory

(Landerer, 2021c) (https://podaac.jpl.nasa.gov/dataset/TELLUS_GRAC_L3_JPL_RL06_LND_v04). The GRACE satellites began orbiting Earth on 17 March 2002; we select the period from 2003 to 2015 for the evaluation. There are 14 missing values during the period, which were filled with a simple linear interpolation.

The GRACE products experience signal leakages between land and lake grids (Save et al., 2016); and such signal leakage could impact the evaluations of the RFCs that are adjacent to the Great Lakes (Ma et al., 2017). To alleviate the impacts of this, we create the reference TWS for the NC, OH, and NE RFCs as follows: (1) aggregate the GRACE-derived TWS over both the RFC land area and neighboring lakes (lakes Superior, Michigan, and Huron for NC; Erie for OH, and Ontario for NE); and (2) subtract the lake water storage anomaly from the aggregated TWS. The lake water storage is calculated as the product

of the observed water level and the lake area.

The lake water level is an arithmetic average of selected National Oceanic and Atmospheric Administration (NOAA) site observations (https://tidesandcurrents.noaa.gov/stations.html?type=Water+Levels). For Lake Superior, five observation stations were selected: Point Iroquois, Marquette C.G., Ontonagon, Duluth, and Grand Marais. For Lake Michigan, seven

stations were selected: Ludington, Holland, Calumet Harbor, Milwaukee, Kewaunee, Sturgeon Bay Canal, and Port Inland. For Lake Huron, five stations were selected: Lakeport, Harbor Beach, Essexville, Mackinaw City, and De Tour Village. For Lake Erie, eight stations were selected: Buffalo, Sturgeon Point, Erie, Fairport, Cleveland, Marblehead, Toledo, and Fermi Power Plant. And for Lake Ontario, four stations were selected: Cape Vincent, Oswego, Rochester, and Olcott.

The lake area is estimated from the lake boundary data provided by the United States Geological Survey (https://www.sciencebase.gov/catalog/item/530f8a0ee4b0e7e46bd300dd). Only the area within the United States is considered, which is within a 150 km radius from the studied RFCs. The lake areas are calculated as follows: 52441 km$^2$ for Lake Superior within the USA; 57509 km$^2$ for Lake Michigan; 23185 km$^2$ for Lake Huron within the USA; 25494 km$^2$ for Lake Erie; and 18871 km$^2$ for Lake Ontario. Month-to-month variations in lake area are neglected in this study for simplicity.

**3.5.2.    Soil moisture**

We use the daily North American Soil Moisture Database (NASMD) (Quiring et al., 2016) as the reference for the simulated soil moisture ($W_{soil,i}$ in Table 1), similar to previous NLDAS evaluations (Xia et al., 2015a, b). NASMD assembles the soil moisture time series at multiple depths of more than 2200 stations of 24 networks with quality control. We obtained the data from the NASMD website at Texas A&M University (http://soilmoisture.tamu.edu/). The observation depth varies with the

network. We interpolated the observations to the centers of the Noah-MP soil layers, which are 0.05 m, 0.25 m, 0.7 m, and 1.5 m, respectively. The interpolation is performed only when a valid observation is exactly at, or two valid observations exist above and below, the given depth; otherwise, a missing value is given. We excluded the soil layers with more than 50% missing

values to minimize the impacts of missing values on the evaluation, after which 264 $w_{soil,1}$, 214 $w_{soil,2}$, 95 $w_{soil,3}$, and 23 $w_{soil,4}$ valid time series remained. Daily data from 1996 to 2013 were then aggregated into monthly values. For any month, if

less than 10 days of valid data is available, a missing value is assigned.

### 3.5.3.   Snow water equivalent

We use the daily Snow Data Assimilation System (SNODAS) as the reference of SWE ($W_{snow}$ in Table 1). SNODAS is a data assimilation system developed by the NOAA National Weather Service's National Operational Hydrologic Remote Sensing Center. This system aims to provide a physically consistent framework to combine snow modeling and observations from

satellites, airborne platforms, and ground stations (National Operational Hydrologic Remote Sensing Center, 2004). We downloaded the dataset from the National Snow and Ice Data Center website (https://nsidc.org/data/G02158/versions/1). The original spatial resolution is 1 km × 1 km, and we bilinearly interpolated the data to the 0.125° NLDAS grids. SNODAS began on 2003-09-28, and we selected 11 years of whole snow seasons from September 2004 to August 2015. Clow et al. (2012) showed that the SNODAS SWE performs well in the forest areas of the Colorado Rocky Mountains, but performs poorly in

the alpine areas.

### 3.5.4.   Evapotranspiration

We use plot-scale AmeriFlux observations and four gridded products as the reference ET ($E$ in Table 1). The four gridded products are derived from different methods, and a common evaluation period from 1982 to 2015 is selected for this study. The gridded products have different spatial resolutions. We downscaled the data to the NLDAS grids and then aggregated

them to the 12 RFCs.

We select 25 AmeriFlux sites (https://ameriflux.lbl.gov/). AmeriFlux is a network of eddy-covariance systems measuring ecosystem $CO_2$, water, and energy fluxes across the United States. The 25 selected sites were selected because they have the longest observation periods for seven major land cover types (i.e., evergreen forest, deciduous forest, mixed forest, shrubland,

savanna, grassland, and cropland). Figure S1 and Table S1 detail the selected sites. AmeriFlux provides hourly or half-hourly latent heat measurements of the selected sites since 1991, and we calculate ET by dividing the latent heat of water vaporization ($2.5104 \times 10^6$ J kg$^{-1}$). In this study, AmeriFlux serves as the ground truth for the gridded ET products and model estimation. The data have been widely used in LSM evaluations (Cai et al., 2014b; Zhang et al., 2020). We aggregated the original hourly or half-hourly data into daily and then monthly values. In the process of aggregation, if there was less than eight valid hours

in a day, a missing day was marked; if there was less than ten valid days in a month, a missing month was assigned; if there was less than 50% valid months, the time series was dropped.



The first gridded ET product is FLUXNET Multi-Tree Ensemble (MTE) (Jung et al., 2009) (https://www.bgc-jena.mpg.de/geodb/projects/Home.php). FLUXNET MTE ET is a monthly data produced from the FLUXNET eddy
covariance measurements, remote sensing, and meteorological data using the multi-tree ensemble statistical method (Jung et al., 2009). The product is widely used in LSM evaluations (Cai et al., 2014b; Ma et al., 2017; Xia et al., 2016; Jung et al., 2019; Fang et al., 2020; Zhang et al., 2020; Pan et al., 2020). FLUXNET MTE ET has a spatial resolution of $0.5° \times 0.5°$. We remap the data to the $0.125° \times 0.125°$ NLDAS grids with a first-order conservative method.

The second gridded ET product is the Global Land Evaporation Amsterdam Model (GLEAM), version 3.3a (https://www.gleam.eu/), which is another widely used ET product (Xu et al., 2019). GLEAM estimates transpiration, canopy evaporation, soil evaporation, open-water evaporation, and sublimation separately, and then sums them as ET. The method aims to maximize the utilization of satellite information. The product estimates monthly ET at a spatial resolution of $0.25° \times 0.25°$. We bilinearly interpolated the data to the NLDAS grids.


The third gridded ET product was developed by Ma and Szilagyi (2019), based on Complementary Relationship (CR) method (https://digitalcommons.unl.edu/natrespapers/986/). Regional ET is estimated as a complementary function (Szilagyi et al., 2017) of maximum attainable ET and wet environment ET. The product requires only commonly available meteorological data but has shown reasonable performance in describing the annual ET cycle, linear trends, and interannual anomaly (Ma et
al., 2019; Ma and Szilagyi, 2019). This CR-based product has a spatial resolution of 4 km, and we used local area averaging to interpolate the data to the NLDAS grids.

We evaluated the above three gridded datasets at 25 AmeriFlux sites using three skill metrics—namely $R$, TSS, and nuRMSE. We selected the grid points closest to these 25 sites for evaluation. The evaluation results are shown in Tables S2–S4. All three
datasets can capture the annual ET cycle well (average TSSs above 0.8). Among them, the FLUXNET MTE ET performs the best. However, all three datasets cannot capture the interannual ET anomaly well (average TSSs below 0.6). Among them, the GLEAM ET performs the best.

### 3.5.5.  NLDAS ensemble

We used three NLDAS-2 models—namely Noah-2.8, VIC-4.0.3, and Mosaic, as the benchmark of the Noah-MP ensemble.
Their outputs can be publicly downloaded from the NASA Goddard Earth Sciences Data and Information Services Center (https://disc.gsfc.nasa.gov/datasets?keywords=NLDAS). More information on the NLDAS-2 models, the forcing and static datasets, and simulation settings can be found in Xia et al. (2012b, a) and Section 2.1 of Fei et al. (2021). The NLDAS-2 datasets have been proven to perform soundly for regional hydrological simulations (Xia et al., 2012b, a, 2016, 2015a, b) and are widely selected for LSM comparisons (Cai et al., 2014a; Fei et al., 2021; Cai et al., 2014b).





## 4. Results and discussion

In this section, we begin in in Section 4.1 by quantifying the spread among the members of the Noah-MP ensemble. Then, Section 4.2 evaluates the estimated TWS anomaly (TWSA), SWE, soil moisture, and ET in comparison with the NLDAS ensemble. More results on runoff can be found in Zheng et al. (2020) and Fei et al. (2021).

### 4.1. Spread among the ensemble members

All the dataset variables are aggregated across CONUS and the spread among the 48 Noah-MP configurations is calculated, as shown in Table 2. Among the variables, runoff (including surface and subsurface components) shows the largest spread, which is comparable to that estimated in GSWP-2. The spread of ET and that of TWS are significantly smaller than those observed in GSWP-2. The high consistency among the Noah-MP configurations could be a sign of the limited sampling of available process parameterizations, but also could be a result of continuous model improvements. ET might be the former case, whereas TWS is likely the latter. The spread in surface soil moisture is marginal and increases significantly in the deep layers. The surface soil moisture is controlled tightly by the atmospheric forcings, whereas the spread of the sub-surface soil moisture hints at the complex interplay among various land surface processes (e.g., root water uptake and subsurface runoff) (Koster, 2015). The constraints of the atmospheric forcing are also obvious for the interannual anomaly, resulting in a relatively smaller ensemble spread. LSM parameterizations play a major role in estimating annual cycles, and the spread in the annual cycle is dominant for the overall ensemble spread of runoff and soil moisture.

### 4.2. Evaluation with observations

#### 4.2.1. Terrestrial water storage anomaly

Figure 1 shows the annual cycle of the TWSA estimated from GRACE, Noah-MP, and NLDAS. Figure 2 presents the TSS. In the 12 RFCs over CONUS, the TWSA peaks in spring, declines rapidly in summer, reaches a minimum in autumn, and recovers in winter. In terms of the timing of the peak and trough, Noah-MP and the NLDAS models perform similarly. In terms of the amplitude of variation, Noah-MP generally produces higher values in all RFCs. Previous studies have attributed this difference to the inclusion of a bucket groundwater component in Noah-MP (Cai et al., 2014a; Ma et al., 2017). However, we found the Noah-MP configurations without a groundwater component can still produce a higher amplitude, especially considering the structural similarity between Noah-MP and Noah. Further investigation of the model difference is necessary.

Figure 2 shows the Taylor diagram for the annual cycle. The Noah-MP configurations generally outperform the NLDAS models in most of the RFCs, which results in a superior ensemble mean performance, as shown in Table S5. Detailed examination of the TSS reveals that Noah-MP and NLDAS have similar correlation coefficients. Their difference is manifested



in the modeled standard deviation (i.e., the amplitude of variation). In NE, MA, NW, and CN, Noah-MP underperforms
compared with NLDAS, mainly due to underestimating the standard deviation. However, the interpretation of the
underestimation is multifaceted. First, they are coastal RFCs, and the GRACE data at the coast could be contaminated by the
leaked signal from the ocean, producing a low temporal variability (Cai et al., 2014a). Second, there are strong human activities
in CN, perturbating the water storage of the aquifers at the deeper levels. Noah-MP and the NLDAS models do not consider
the groundwater variations at such depths, resulting in an underestimation. For the same reason, the underestimation is also
pronounced in AB, which is home to the Ogallala Aquifer and has strong human activities. Third, Noah-MP does not include
a lake module. The estimates do not include the variations in lake water storage.

Figure 3 shows the Taylor diagram for the interannual anomaly. Compared with the annual cycle (Figure 2), both the Noah-
MP configurations and the NLDAS models degrade significantly. Noah-MP still shows better skill than NLDAS in most of
the RFCs. However, its superiority is marginal. In three RFCs—namely NE, MA, and WG, Noah-MP underperforms NLDAS,
and the underperformance corresponds to an underestimated standard deviation. Similar to the annual cycle, we are of the
opinion that possible reasons could be an exclusion of lakes in Noah-MP and the coastal signal leakage in the GRACE estimates.

### 4.2.2.   Soil moisture

Figure 4 presents the time series of the surface (0–0.1 m) and root-zone (0–1.0 m) soil moisture in NC, NW, AB, WG, and
CB. These RFCs were selected as they have more than 10 valid sites, and the time series is averaged across the sites.
Corresponding TSSs are provided in Table S6. The 48 Noah-MP configurations are consistent in estimating the surface soil
moisture, and the spread is remarkably smaller than that among the three NLDAS models. The spread among the Noah-MP
configurations increases significantly from the surface (Figure 4e) to the root zone (Figure 4k) for the soil moisture in AB. The
ensemble spread in the root-zone soil moisture reflects the difference in modeling root-water uptake for plant transpiration and
soil-bottom drainage as described in Section 2.2. Further investigation (Figures S2–S5) shows that, in the deep soil layers (the
third and fourth layer), Noah-MP has comparable or greater spread than NLDAS. In the RFCs and soil layers examined in
Figure 4, Noah-MP outperforms the NLDAS models (Table S6), which is consistent with previous evaluations (Cai et al.,
2014a). After further comparing Noah-MP and Noah, it is interesting to note that Noah-MP and Noah perform similarly in AB
(Figures 4e and 4f) and WG (Figures 4g and 4h) but are different in NC (Figures 4a and 4b), NW (Figures 4c and 4d), and CB
(Figures 4i and 4j). The similarity in AB and WG is reasonable since the two models have similar soil layer structures and
parameterizations. The dissimilarity in NC, NW, and CB is most pronounced in winter. It could result from the different snow
parameterizations in Noah-MP and Noah, which is investigated in Section 4.2.3.

Figures 5 and 6 compare the TSS between the NLDAS and Noah-MP ensemble mean at each NAMSD site for the annual
cycle and interannual anomaly. The comparison varies significantly with site and soil layer depth, revealing two major patterns.



First, NDLAS outperforms Noah-MP in AB and CB. In these two RFCs, as individual NLDAS models do not show superior performance over Noah-MP (Figure 4), the high skill of the ensemble mean could be a result of a high ensemble skill gain (Fei et al., 2021) related to the diversity among the NLDAS models (Guo et al., 2007; Xia et al., 2015a). On the other hand, Noah-MP has a low ensemble spread. The good performance of individual Noah-MP configurations does not turn into a higher skill

of the ensemble mean owing to a lack of diversity in soil structures and parameterizations. Second, in NC, OH, and MA, NLDAS underperforms Noah-MP, and the low performance of NLDAS is related to the anomaly in winter-time soil moisture (Figure S2). The anomaly suggests that the NLDAS models have difficulty in modeling snow and snow-soil moisture interactions (refer to Section 4.2.3 for more information). On the other hand, Noah-MP has a better snow module, leading to a high soil moisture estimation skill.


To maximize the utilization of the NLDAS model diversity and Noah-MP physics improvements, we combine the Noah-MP ensemble mean and the three NLDAS models. The right-hand columns of Figures 5 and 6 show that the four estimates' arithmetic average outperforms the three-model NLDAS ensemble mean at almost every NASMD site, suggesting added value in the Noah-MP data.

### 525    4.2.3.   Snow water equivalent

Figure 7a presents the spatial patterns of the 11-year average (2004-09 to 2015-08) SWE ($W_{snow}$) from SNODAS. Over CONUS, snow is mainly distributed in the northeast (NE, NC, OH, and MA) and on the mountains of the west (the Cascade Mountains, Rocky Mountains, Sierra Nevada in NW, AB, MB, WG, CN, and CB). Figure 7b (7c) shows the geographical difference between the Noah-MP (NLDAS) ensemble mean and SNOWDAS. Both Noah-MP and NLDAS exhibit a

considerable underestimation in most areas of CONUS. However, the underestimation of Noah-MP is smaller in areas where snow is thick (e.g., NE and the Cascade Mountains). Consequently, Noah-MP captures the spatial patterns better than NLDAS, with a spatial correlation of 0.67 versus 0.31.

Figure 8 compares the annual cycle estimated from SNODAS, NLDAS, and Noah-MP. The annual cycle in the 12 RFCs

exhibits a similar pattern: it accumulates in winter, peaks in spring, and melts from late spring to summer. The snow season in the northeastern RFCs (i.e., NE, MA, OH, and NC) spans from October to May, whereas the snow season is longer in the mountainous RFCs of the west (i.e., NW, AB, MB, WG, CN, and CB), lasting to June. From the comparison between Noah-MP and NLDAS, we make three observations. First, the NLDAS models underestimate the SWE in all RFCs. Among the three NLDAS models, Noah performs the best in the northeast (e.g., NE and MA), whereas VIC shows some advantages in the

mountains (e.g., NW, CN, and CB). In the northeast, possible reasons for the Noah superiority include the careful consideration of the surface energy balance over flat terrain, whereas in the mountains, the elevation bands of VIC can better capture the spatial heterogeneity. Second, Noah-MP is close to SNODAS in the northeastern RFCs (e.g., NE, MA, OH, and NC), which



has flat terrain and thick snow. This could be a consequence of the multi-layer snow module (Niu et al., 2011) and consideration of the impacts of surface energy balance on snow accumulation/ablation. However, in RFCs such as LM, AB, MB, and WG,

Noah-MP does not show clear superiority, and one possible reason is that the snow albedo parameterization is biased when the snow is shallow, as discussed in Dang et al. (2019) and Wang et al. (2020). Third, the spread of the Noah-MP ensemble is small. The selected parameterization configurations do not differ much in terms of modeling the SWE. The Noah-MP configurations should be averaged as a single value when combined with other estimates such as NLDAS. The above three observations suggest that a high spatial resolution (or elevation bands in coarse-resolution models), a multi-layer snow model,

and improved shallow snow albedo parameterizations would be critical for accurate SWE estimations.

Figure 9 shows the TSS of the NLDAS and Noah-MP ensembles in estimating the annual cycle and interannual anomaly. Table S7 summarizes the skill scores for the 12 RFCs. The annual cycle and interannual anomaly exhibit similar spatial patterns. NLDAS performs well in the northern part of CONUS (e.g., NE, MA, NC, and NW), with TSSs higher than 0.7. The

performance in the southern part of CONUS is low. However, snow occurs sparsely in these areas and may not be a significant part of the terrestrial water cycle. In comparison with NLDAS, Noah-MP shows superiority in areas with thick snow (e.g., in the northeast and northwest of CONUS), but underperforms in areas with shallow snow (e.g., in central CONUS). As discussed in the previous paragraph, the Noah-MP snow module is appropriate when snow is thick, but needs improvement when snow is shallow. We further averaged the 48 Noah-MP configurations and added their average to the three-model NLDAS ensemble.

Figures 9e and 9f show that the four-estimate ensemble mean outperforms the three-model NLDAS ensemble mean in nearly all areas of CONUS, again proving the added value of the data provided in this paper.

### 4.2.4.    Evapotranspiration

Figure 10 compares the annual cycle estimated from FLUXNET MTE, NLDAS, and Noah-MP in the 12 RFCs. We choose FLUXNET MTE as the reference here since its performance is superior when compared to AmeriFlux (Tables S2–S4). In all

12 RFCs, ET peaks during summer and is lowest during winter. Noah-MP successfully captures the timing of the peak in humid RFCs (i.e., NE, MA, OH, LM, SE, NC, and NW) but shows a one-month lead in a few semi-arid and arid RFCs (i.e., MB, WG, and CN).  The average of the three NLDAS models better reproduces the timing of the peak, but the models differ from each other significantly. Among the Noah-MP and NLDAS ensembles, VIC and Mosaic are notably different. VIC exhibits a systematic underestimation, while Mosaic shows an overall overestimation. The 48 Noah-MP configurations and

Noah perform closely during autumn and winter, whereas their differences are pronounced during spring and summer. During spring and summer, Noah is the closest to FLUXNET MTE in most RFCs except NE and MA, whereas Noah-MP constantly overestimates the ET in all RFCs.





The overestimation of Noah-MP was investigated by separately comparing the three components of ET (i.e., transpiration,

canopy evaporation, and ground evaporation) with GLEAM (Figure S6). Figure 11 shows the overestimation of total ET is

closely linked to the overestimation of ground evaporation, which could be partially attributable to the overly high roughness

length for heat and water, as described in Sections 2.2.8 and 2.2.9. Besides, the lack of a litter layer (Decker et al., 2017) in

Noah-MP could also play a part.

Figure 12 evaluates Noah-MP and NLDAS using the 25 AmeriFlux sites. The NLDAS ensemble mean outperforms the Noah-

MP ensemble mean for the annual cycle, and this outperformance results from two causes. First, an NLDAS member, Noah,

performs the closest to the observations, as shown in Figures 12b and 10. Second, the three NLDAS models are remarkably

different from each other. The diversity of the ensemble gives a higher skill gain by combining them, as shown by the difference

between the ensemble mean skill (Figures 12a) and the median TSS (Figures 12b). On the other hand, the Noah-MP

configurations are too similar to each other, and all have a positive bias (Figure 10). However, for the interannual anomaly,

the Noah-MP ensemble mean slightly outperforms the NLDAS ensemble mean (Figure 12c). Figure 12d shows that the Noah-

MP configurations marginally outperform the NLDAS models. Among the NLDAS models, VIC performs the best, and Noah

does not exhibit the same superiority shown in the annual cycle. The difference between the NLDAS ensemble mean

performance (Figure 12c) and median performance (Figure 12d) is marginal, suggesting that the NLDAS ensemble skill gains

are not notable for the interannual anomaly.

Figure 13 examines the ensemble spread of Noah-MP and NLDAS. The ensemble spread is normalized by the temporal

variability calculated using the FLUXNET MTE ET. NLDAS has a significant spread in the southeast and west in all seasons,

while spring shows the largest value. As seen in Figure 10, the NLDAS ensemble spread mainly reflects the differences

between VIC and Mosaic. The Noah-MP ensemble has a notably smaller spread than NLDAS. The Noah-MP ensemble spread

is manifested in spring and summer in the southeastern (SE, LM, and WG) and western (CN, CB, and NW) RFCs.

We can decompose the Noah-MP ensemble spread and pinpoint the dominant process using Sobol' sensitivity analysis (Zheng

et al., 2019). Figure 14 delineates the Sobol's total sensitivity index of total ET to the four processes described in Section 2.2.

In spring and summer, for the regions where the Noah-MP configurations show significant spread (SE, LM, WG, CB, CN, and

NW) (Figure 13), ET is most sensitive to the parameterization of stomatal conductance (Figures 14e and 14i) and then to the

$\beta$-factor (Figures 14f and 14j). However, for regions with positive biases (NC, OH, and LM, as shown in Figures 11b and 11c),

the Noah-MP estimation is more sensitive to the turbulence parameterizations (Figures 14c and 14g). During autumn and

winter, the parameterizations of stomatal conductance (Figure 14m and 14q) and $\beta$-factor (Figures 14n and 14r) still have

significant impacts on the estimation of ET, and these impacts could be a result of the "memory" of TWS (Zheng et al., 2019).

Besides these two processes, the runoff parameterization is dominant during autumn in the east (Figure 14p), and the turbulence

parameterization is dominant during winter (Figure 14s).



## 5.    Data availability

The dataset is freely available for download from the Zenodo online repository at https://doi.org/10.5281/zenodo.7109816
(Zheng et al., 2022). The dataset (along with datasets on which it is based) is subject to a Creative Commons BY (attribution) license agreement (https://creativecommons.org/licenses, last access: 2021-08-16).

## 6.    Conclusions

This study involved the construction of an ensemble containing 48 perturbed-physics configurations of Noah-MP with a spatial resolution of 1/8° to estimate the TWB over CONUS from 1980 to 2015. This Noah-MP multi-physics ensemble features an
enrichment of the original four NLDAS-2 models and brings convenience for multi-model comparison. The dataset has already been used in the monitoring of groundwater storage change (Rateb et al., 2020), the analysis of LSM parameterization sensitivity (Zheng et al., 2019), the development of model evaluation method (Zheng et al., 2020), and hydrological ensemble simulations (Fei et al., 2021). This paper details the Noah-MP parameterizations employed and evaluates the estimated TWSA, soil moisture, SWE, and ET in comparison with the NLDAS ensemble.


The Noah-MP estimates are closer to the reference than NLDAS for TWS, SWE, and soil moisture. The multi-layer snow module of Noah-MP shows superiority not only for estimating SWE but also for winter-time soil moisture. The Noah-MP ensemble complements the NLDAS multi-model ensemble well; adding the Noah-MP estimates can consistently improve the NLDAS estimates of the above variables in most areas of CONUS. For ET, Noah-MP outperforms NLDAS for the interannual
anomaly but performs poorly for the annual cycle. For the annual cycle, there is a systematic overestimation in spring and summer over OH, NC, and LM. On the basis of a Sobol' sensitivity analysis of the Noah-MP estimation, the biases could be mainly related to the parameterization of turbulence, the code of which is inconsistent with the original literature and overestimates the roughness length of heat and water vapor.

The study shows that the Noah-MP perturbed-physics ensemble is useful not only in improving water budget estimations over CONUS, but also in better identifying model deficiencies. The Noah-MP ensemble has the following shortcomings to be improved: (1) human activities and lakes are essential for reproducing the observed TWS but are still missed in Noah-MP; (2) accurate SWE estimation requires a multi-layer snow module, a high spatial resolution, and better shallow snow albedo parameterizations, whereas Noah-MP has to be improved for the latter two; and (3) turbulence parameterizations in Noah-MP
have to be scrutinized, especially considering some critical departures between the code implementation and literature.



**Author contributions**

ZLY initiated and funded the study. HZ conducted the simulation and generated the data. WF analyzed the data and created the figures. WYW, PL, and JW contributed to the validation of the data. All authors contributed to creating the dataset and drafting the manuscript.

**Competing interests**

The authors declare that they have no conflict of interest.

**Disclaimer**

The data are provided as is with no warranties.

**Acknowledgments**

This work was financially supported by the National Natural Science Foundation of China (grants 42075165, 41375088, and 41605062).

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





**Table 1:** Dataset variables

| Symbol | Units | Description |
|---|---|---|
| | | *surface water budget* |
| $E$ | kg m$^{-2}$ s$^{-1}$ | total evaporation |
| $E_{can}$ | kg m$^{-2}$ s$^{-1}$ | evaporation of canopy interception |
| $E_{gnd}$ | kg m$^{-2}$ s$^{-1}$ | direct evaporation from the ground |
| $E_{tran}$ | kg m$^{-2}$ s$^{-1}$ | transpiration |
| $R$ | kg m$^{-2}$ s$^{-1}$ | total runoff |
| $R_{srf}$ | kg m$^{-2}$ s$^{-1}$ | surface runoff |
| $R_{sub}$ | kg m$^{-2}$ s$^{-1}$ | subsurface runoff |
| $W$ | kg m$^{-2}$ | terrestrial water storage |
| $W_{snow}$ | kg m$^{-2}$ | snow water equivalent |
| $W_{gw}$ | kg m$^{-2}$ | groundwater storage |
| $w_{soil,i}$ | m$^3$ m$^{-3}$ | volumetric soil water content |
| $z_{snow}$ | m | snow depth |
| | | *auxiliary variables* |
| $X$ | - | land-water mask (1 for land, 2 for water) |



**Table 2:** The ensemble spread, temporal variability, and rating of different water budget components. $\sigma_{lss\_ancy}$, $\sigma_{lss\_anom}$, and $\sigma_{lss\_total}$ denote the spread of the 48 Noah-MP configurations in simulating the multi-year averaged annual cycle, interannual anomaly, and total 36-year monthly time series, respectively. $\sigma_{Ancy}$, $\sigma_{Anom}$, and $\sigma_{Total}$ denote the temporal variability of the annual cycle, interannual anomaly, and the 36-year monthly, respectively. $R_{ancy}$, $R_{anom}$, and $R_{total}$ denote the rating of the three above-mentioned time scales based on the normalized ensemble spread, respectively $W'$ ($W'_{gw}$) denotes the terrestrial water storage (groundwater) anomaly (kg m$^{-2}$), whereas $\Delta W$ ($\Delta W_{gw}$) denotes the monthly terrestrial water storage (groundwater) change (kg m$^{-2}$ s$^{-1}$).

| Variables | Ensemble spread | | | Temporal variability | | | R | | | Rating | | |
|---|---|---|---|---|---|---|---|---|---|---|---|---|
| | $\sigma_{lss\_ancy}$ | $\sigma_{lss\_anom}$ | $\sigma_{lss\_total}$ | $\sigma_{ancy}$ | $\sigma_{anom}$ | $\sigma_{total}$ | $R_{ancy}$ | $R_{anom}$ | $R_{total}$ | ancy | anom | total |
| $E$ (kg m$^{-2}$ s$^{-1}$) | 1.1938 ×10$^{-6}$ | 2.6612 ×10$^{-7}$ | 1.2223 ×10$^{-6}$ | 1.1764 ×10$^{-5}$ | 1.1842 ×10$^{-6}$ | 1.1823 ×10$^{-5}$ | 0.1015 | 0.2247 | 0.1034 | A | A | A |
| $E_{can}$ (kg m$^{-2}$ s$^{-1}$) | 2.0698 ×10$^{-7}$ | 4.1702 ×10$^{-8}$ | 2.0699 ×10$^{-7}$ | 1.1083 ×10$^{-6}$ | 3.0769 ×10$^{-7}$ | 1.1502 ×10$^{-6}$ | 0.1868 | 0.1355 | 0.1800 | A | A | A |
| $E_{gnd}$ (kg m$^{-2}$ s$^{-1}$) | 7.3255 ×10$^{-7}$ | 1.4349 ×10$^{-7}$ | 7.4295 ×10$^{-7}$ | 3.6413 ×10$^{-6}$ | 7.7774 ×10$^{-7}$ | 3.7235 ×10$^{-6}$ | 0.2012 | 0.1845 | 0.1995 | A | A | A |
| $E_{tran}$ (kg m$^{-2}$ s$^{-1}$) | 1.3345 ×10$^{-6}$ | 2.0629 ×10$^{-7}$ | 1.3483 ×10$^{-6}$ | 7.8742 ×10$^{-6}$ | 6.6551 ×10$^{-7}$ | 7.9023 ×10$^{-6}$ | 0.1695 | 0.3100 | 0.1706 | A | A | A |
| $R$ (kg m$^{-2}$ s$^{-1}$) | 1.2233 ×10$^{-6}$ | 3.6872 ×10$^{-7}$ | 1.2591 ×10$^{-6}$ | 3.1335 ×10$^{-6}$ | 3.6872 ×10$^{-7}$ | 1.2591 ×10$^{-6}$ | 0.3904 | 0.2061 | 0.3490 | B | A | B |
| $R_{srf}$ (kg m$^{-2}$ s$^{-1}$) | 6.8201 ×10$^{-7}$ | 2.2459 ×10$^{-7}$ | 7.0071 ×10$^{-7}$ | 6.9029 ×10$^{-7}$ | 5.7802 ×10$^{-7}$ | 9.0033 ×10$^{-7}$ | 0.9880 | 0.3885 | 0.7783 | B | B | B |
| $R_{sub}$ (kg m$^{-2}$ s$^{-1}$) | 1.0013 ×10$^{-6}$ | 3.4277 ×10$^{-7}$ | 1.0389 ×10$^{-6}$ | 2.4692 ×10$^{-6}$ | 1.3121 ×10$^{-6}$ | 2.7962 ×10$^{-6}$ | 0.4055 | 0.2612 | 0.3712 | B | A | B |
| $W'$ (kg m$^{-2}$) | 5.5732 | 3.4333 | 6.4796 | 44.9508 | 18.0598 | 48.4430 | 0.1240 | 0.1901 | 0.1338 | A | A | A |
| $\Delta W$ (kg m$^{-2}$ s$^{-1}$) | 2.8444 | 0.8990 | 2.9549 | 22.5839 | 5.9302 | 23.3270 | 0.1259 | 0.1516 | 0.1267 | A | A | A |
| $W'_{gw}$ (kg m$^{-2}$) | 0.6713 | 0.8549 | 1.0856 | 8.1760 | 6.1079 | 10.1988 | 0.0821 | 0.1400 | 0.1064 | A | A | A |
| $\Delta W_{gw}$ (kg m$^{-2}$ s$^{-1}$) | 0.3742 | 0.3005 | 0.4812 | 4.1177 | 1.6318 | 4.4243 | 0.0909 | 0.1842 | 0.1088 | A | A | A |
| $W_{snow}$ (kg m$^{-2}$) | 0.1254 | 0.1273 | 0.1726 | 7.5644 | 3.8036 | 8.4669 | 0.0166 | 0.0335 | 0.0204 | A | A | A |
| $w_{soil,1}$ (m$^3$ m$^{-3}$) | 0.0066 | 0.0009 | 0.0067 | 0.0242 | 0.0102 | 0.0262 | 0.2726 | 0.0933 | 0.2537 | A | A | A |
| $w_{soil,2}$ (m$^3$ m$^{-3}$) | 0.0084 | 0.0012 | 0.0085 | 0.0194 | 0.0081 | 0.0210 | 0.4337 | 0.1522 | 0.4040 | B | A | B |
| $w_{soil,3}$ (m$^3$ m$^{-3}$) | 0.0119 | 0.0018 | 0.0121 | 0.0227 | 0.0088 | 0.0244 | 0.5251 | 0.2080 | 0.4958 | B | A | B |
| $w_{soil,4}$ (m$^3$ m$^{-3}$) | 0.0146 | 0.0018 | 0.0147 | 0.0160 | 0.0071 | 0.0175 | 0.9110 | 0.2594 | 0.8394 | B | A | B |



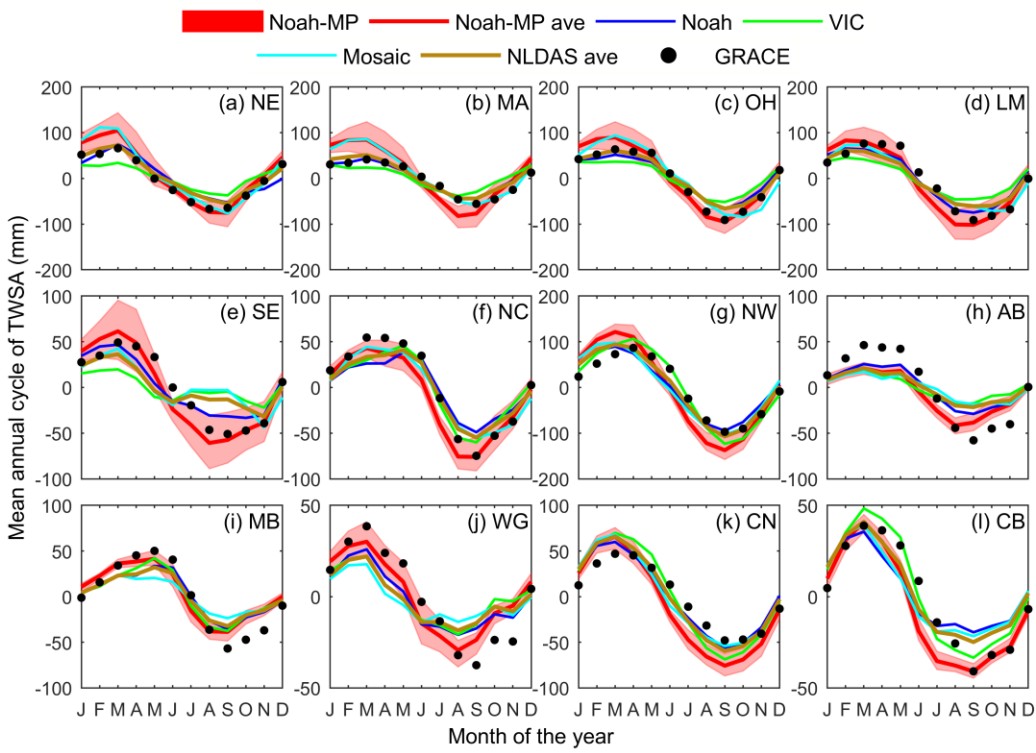

**Figure 1:** Annual cycle from modeled and GRACE-derived TWSA in 12 RFCs for the period 2003–2015. TWSA is calculated from TWS by removing the 13-year average (2003 to 2015). Black dots denote the GRACE TWSA. The shaded areas denote

the range between the maxima and minima of the 48 Noah-MP estimates. The solid red line denotes the Noah-MP multiphysics ensemble mean. The three NLDAS models (Noah, Mosaic, VIC) and their ensemble mean are denoted by the blue, green, cyan, and dark golden lines, respectively. The 12 RFCs are sorted based on climatic aridity, i.e., the most humid RFC in the top left and the driest RFC in the bottom right.

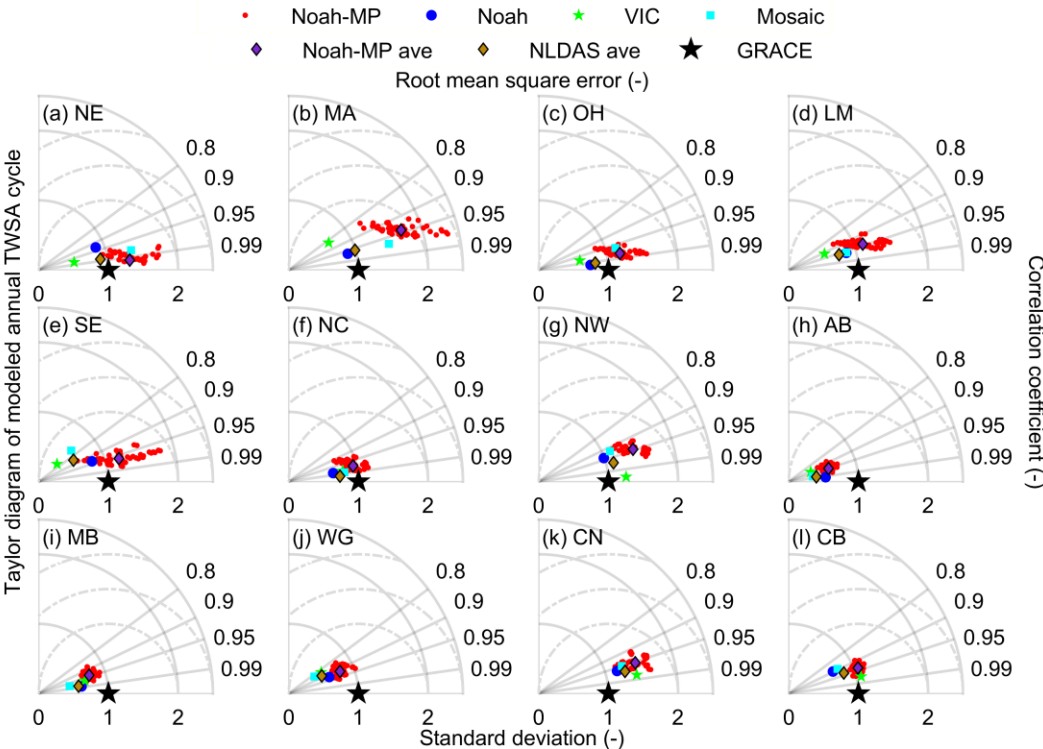

**Figure 2:** Normalized Taylor diagrams showing the performance of the modeled annual TWSA cycle from the 48 Noah-MP ensemble members, which are denoted by the red dots, the three NLDAS models (Noah, VIC, and Mosaic) (blue dot, green star, and cyan square), the Noah-MP ensemble mean (purple diamond), and the NLDAS ensemble mean (dark golden diamond) in each RFC. The black star denotes the observations. The distance between a point of the model simulation to the observations denotes the nuRMSE. The radial lines denote the correlation coefficient, while the distance to the origin along the line denotes the normalized variability.

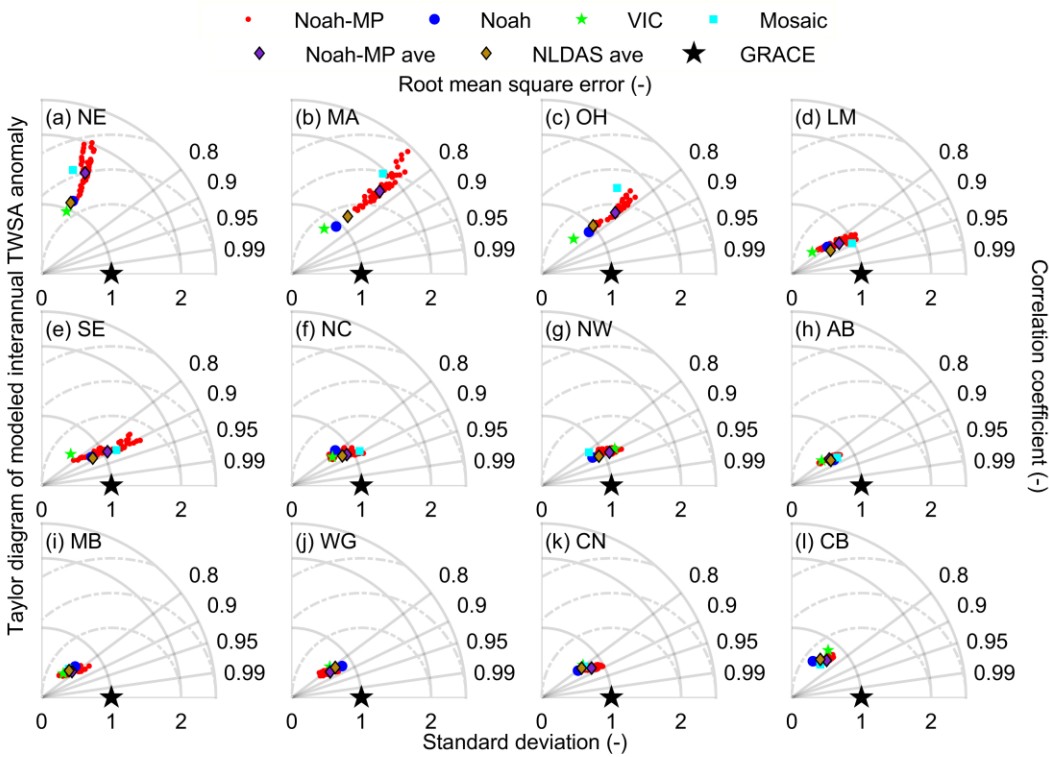

**Figure 3:** As in Figure 2, but for the interannual TWSA anomaly.





**Figure 4:** Monthly soil moisture at 0–0.1 m and 0–1.0 m from the Noah-MP ensemble, the NLDAS models, and the NASMD observations for the period 1996–2013. Only the RFCs with more than 10 observational sites are considered. Black dots denote the NASMD soil moisture observations. The shaded areas denote the range between the maxima and minima of the 48 Noah-MP estimates. The solid red line denotes the Noah-MP multi-physics ensemble mean. The three NLDAS models (Noah, Mosaic, VIC) and their ensemble mean are denoted by the blue, green, cyan, and dark golden lines, respectively. The 12 RFCs are sorted based on climatic aridity.



Earth System Discussions
Open Access Science
Data

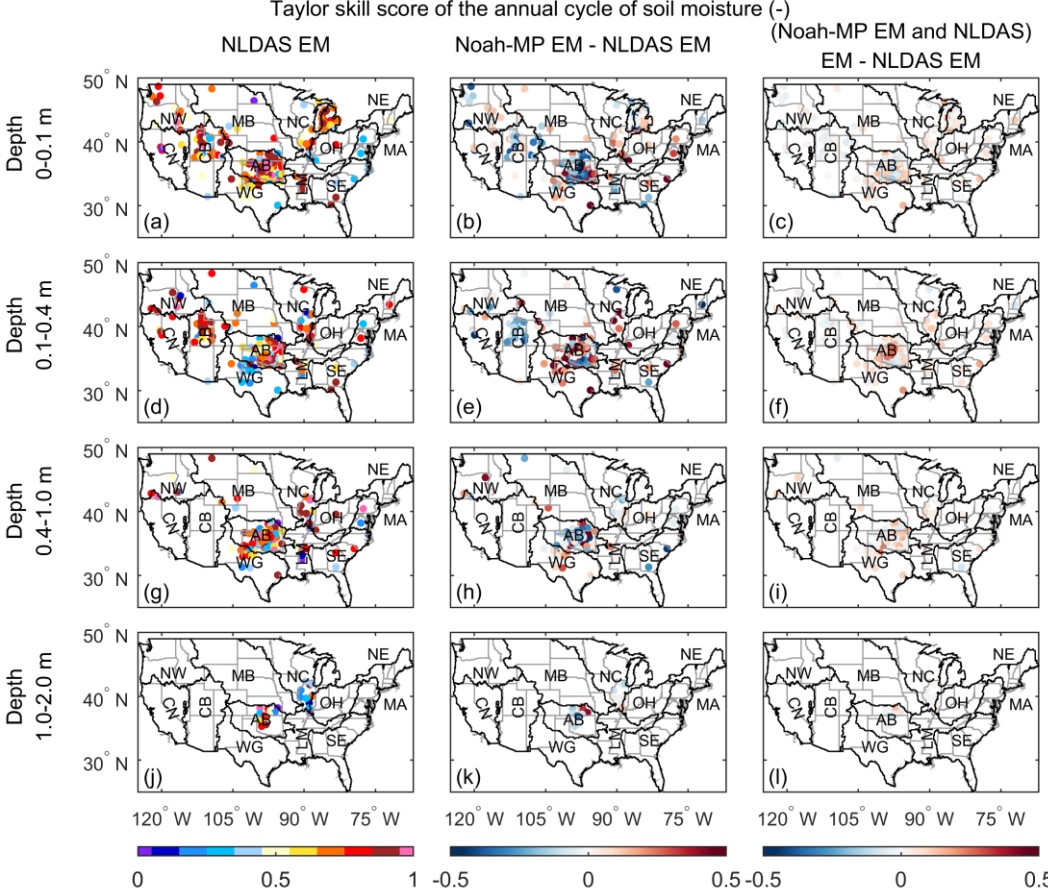

**Figure 5:** TSSs of the NLDAS ensemble mean in simulating the annual cycle of soil moisture (first column) and its performance differences between the Noah-MP ensemble mean (second column), the mean value of the Noah-MP ensemble mean and three NLDAS models (third column) and the NLDAS ensemble mean. The four rows indicate the soil moisture at four different depth ranges (0–0.1 m, 0.1–0.4 m, 0.4–1.0 m, and 1.0–2.0 m). The evaluation period is 1996–2013.



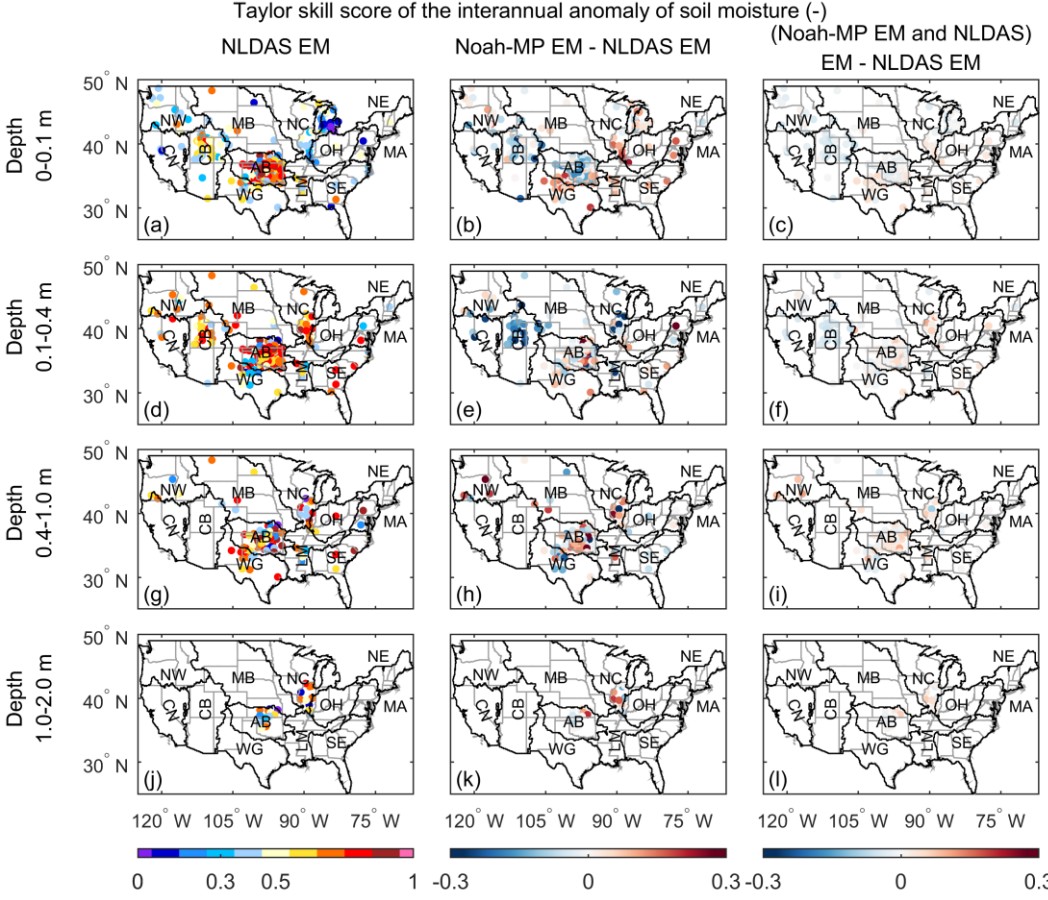

Figure 6: As in Figure 5, but for the interannual anomaly.

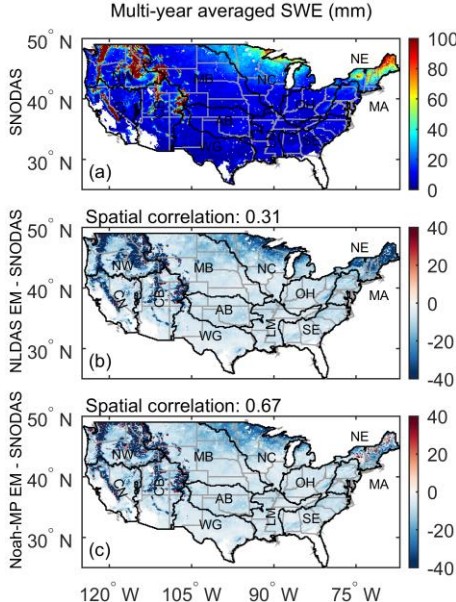

**Figure 7:** Spatial distribution of the 11-year-averaged SNODAS SWE (a); spatial distribution of the multi-year-averaged relative biases between the SNODAS SWE and the NLDAS ensemble mean (b), the Noah-MP ensemble mean (c). The spatial correlation coefficients between the two ensemble means (b, c) and the SNODAS SWE (a) are also presented.



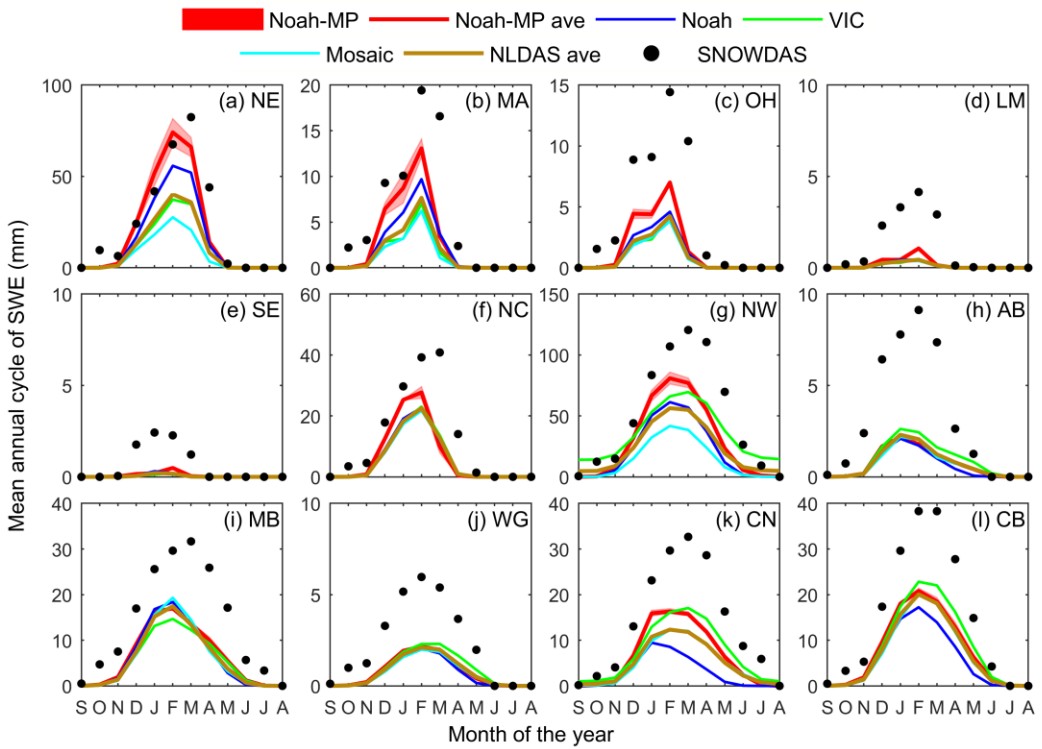

**Figure 8:** As in Figure 1, but for SWE. The evaluation period is September 2004 to August 2015.

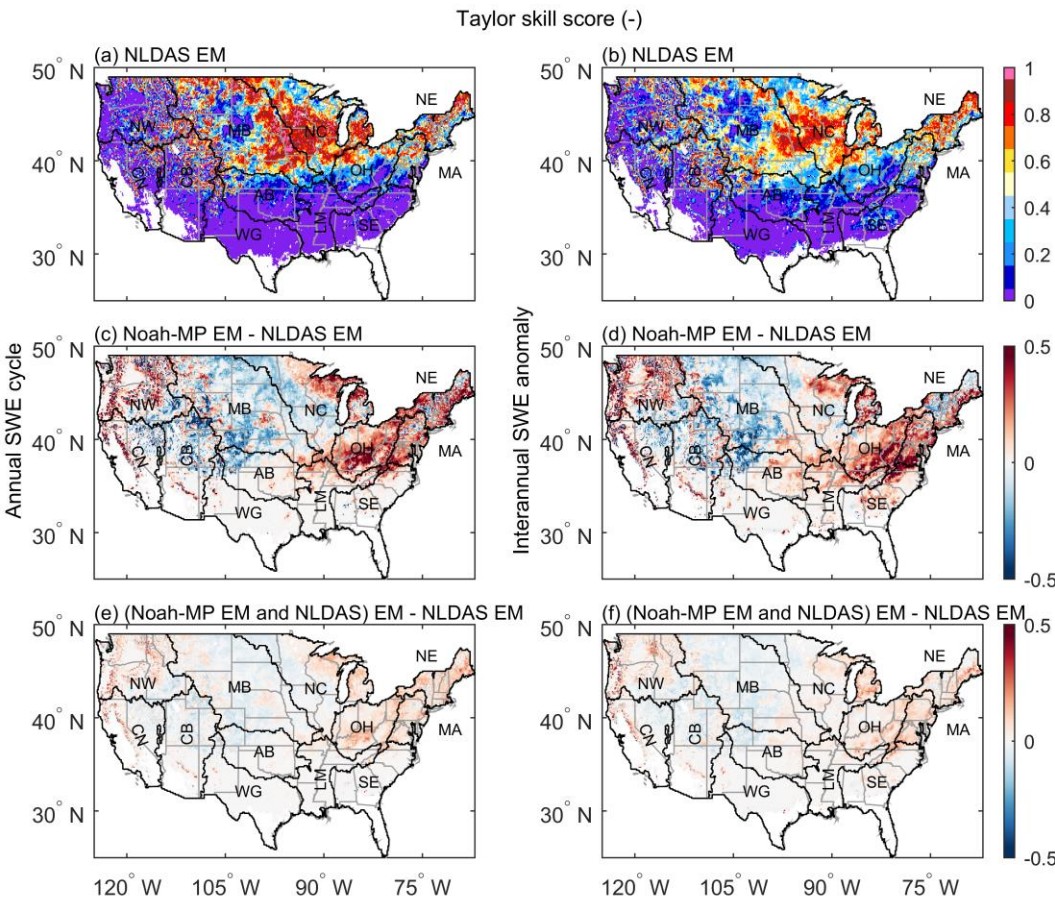

**Figure 9:** TSSs of the NLDAS ensemble mean in simulating the SWE and the performance differences between the Noah-MP ensemble mean (c, d), as well as the mean value of the Noah-MP ensemble mean and three NLDAS models (e, f) and the NLDAS ensemble mean. The first column is for the annual cycle, and the second column is for the interannual anomaly. The evaluation period is 2004–2015.

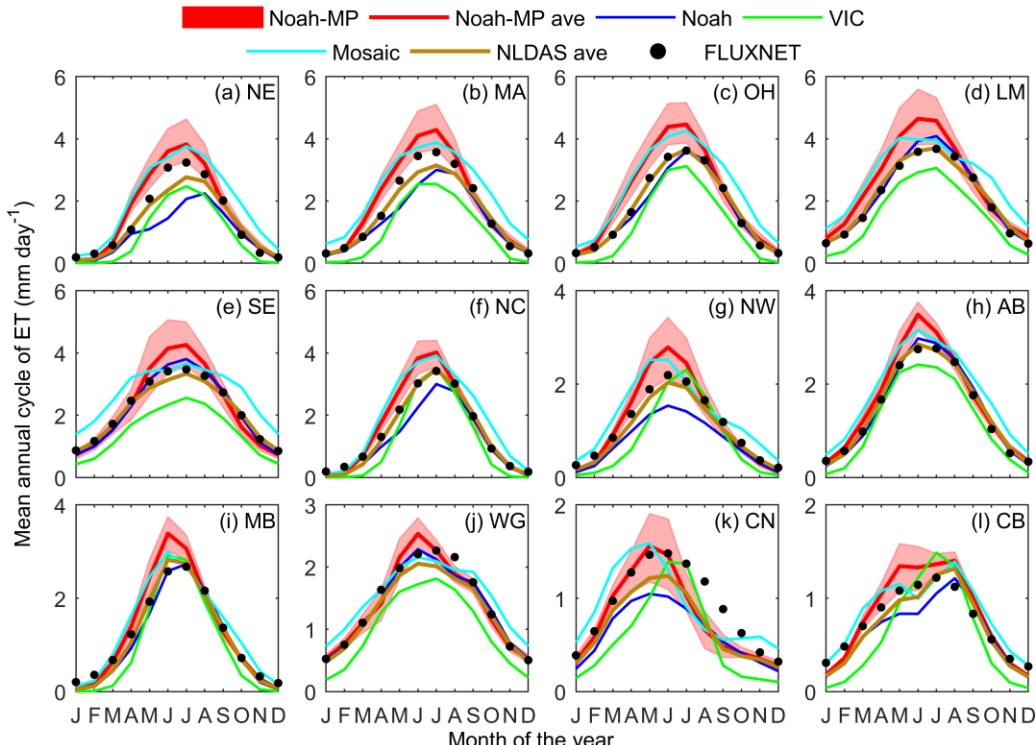

**Figure 10:** As in Figure 1, but for ET. The evaluation period is 1982–2011.

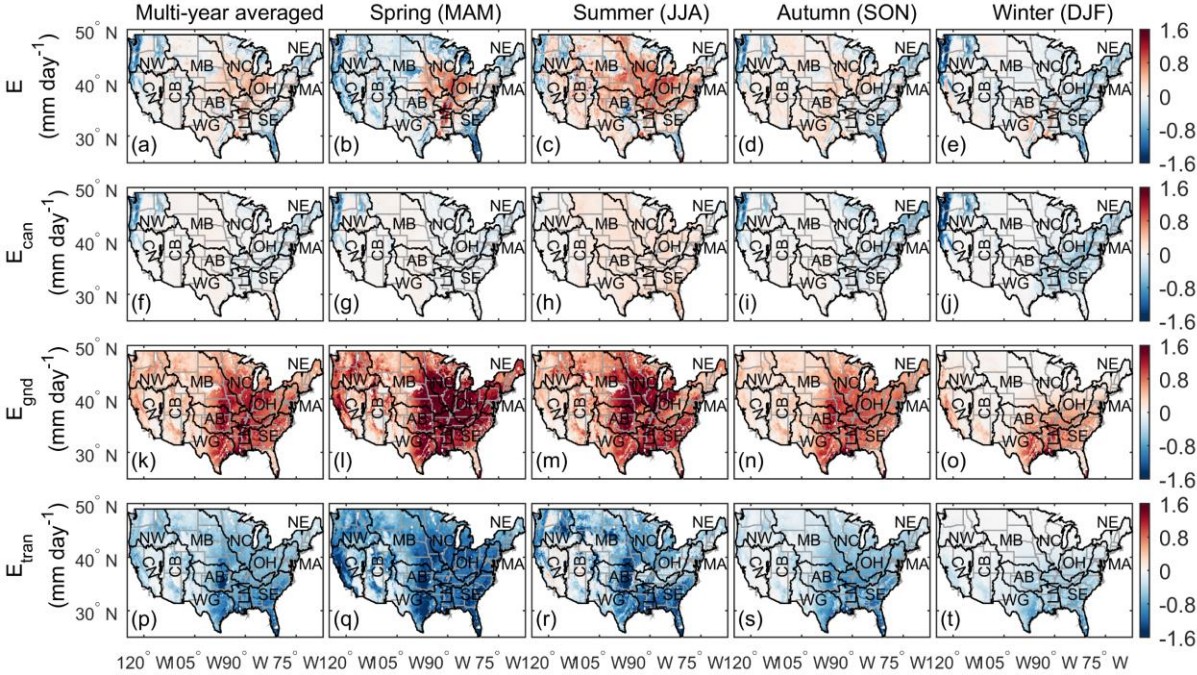

1065

**Figure 11:** Differences between the Noah-MP ensemble mean and GLEAM in total ET ($E$), canopy evaporation ($E_{can}$), ground evaporation ($E_{gnd}$), and transpiration ($E_{tran}$). The units are mm day$^{-1}$.



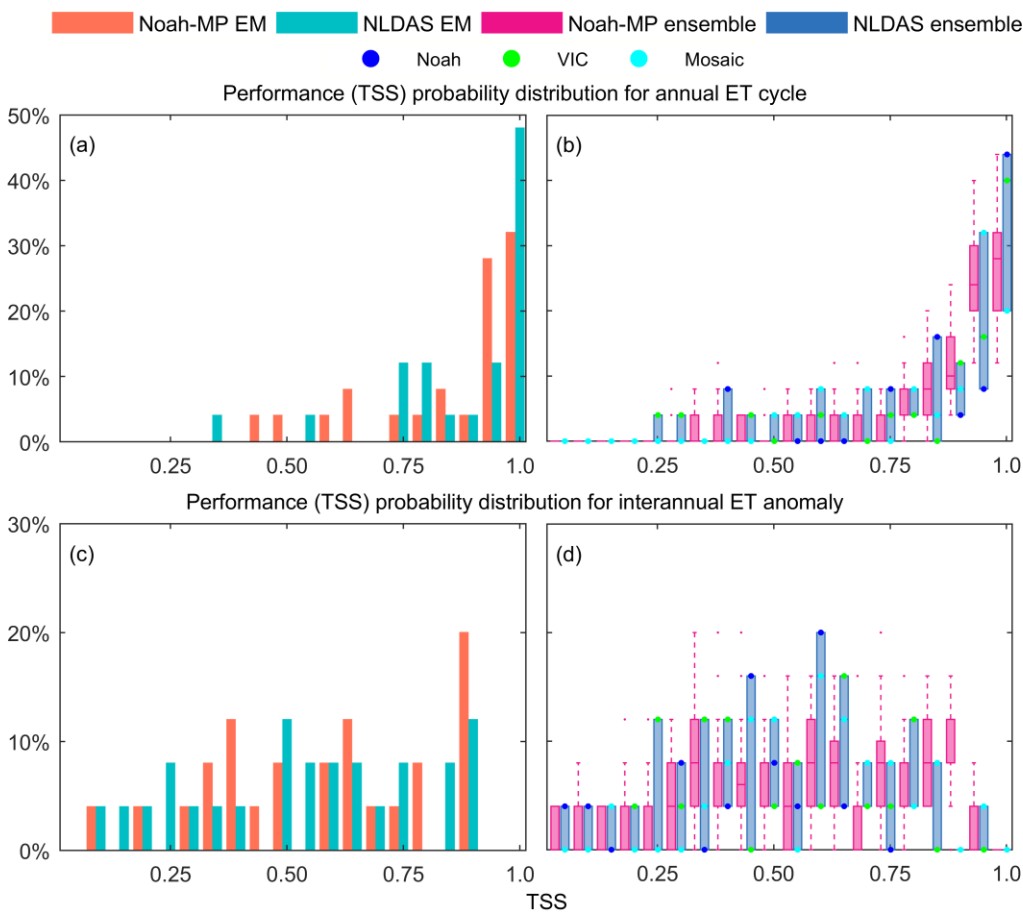

**Figure 12:** TSSs probability distributions of the annual ET cycle (a, b) and interannual ET anomaly (c, d). The orange and cyan bars denote the Noah-MP and NLDAS ensemble means. The magenta and dark blue boxes denote the Noah-MP and NLDAS ensembles. The upper, middle, and lower quantile lines of the magenta boxes show the 75th, 50th, and 25th percentile values of the Noah-MP ensemble. The upper, middle, and lower lines of the dark blue boxes show the three NLDAS models. The blue, green, and cyan dots denote Noah, VIC, and Mosaic, respectively. The evaluation period can be found in Table S1.





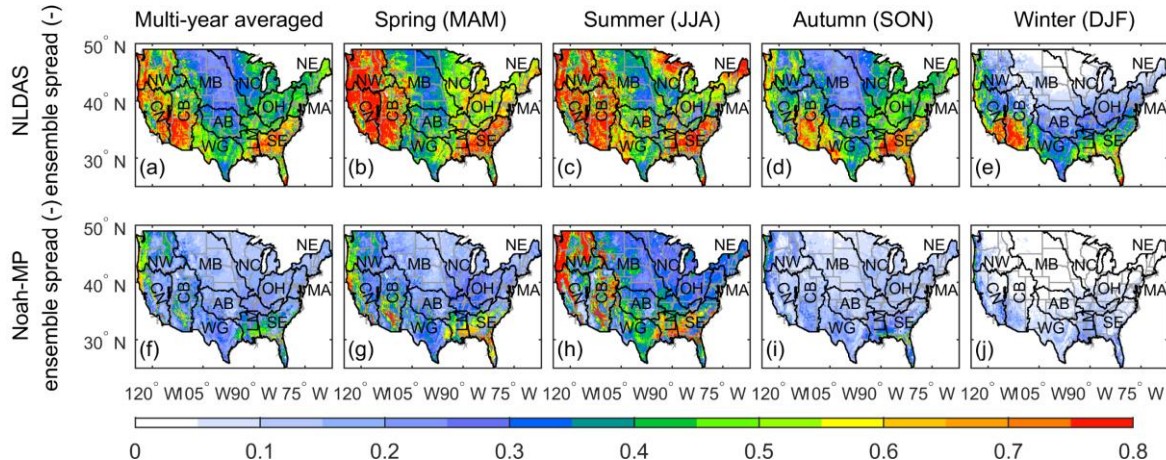

**Figure 13:** Spread of the multi-year averaged annual (first column) and seasonal (spring—MAM, summer—JJA, autumn—SON, winter—DJF) ET (second–fifth columns) from the NLDAS (first row) and Noah-MP (second row) ensembles. The ensemble spread is normalized by the temporal variability of the FLUXNET MTE ET calculated using equation (36).

1080

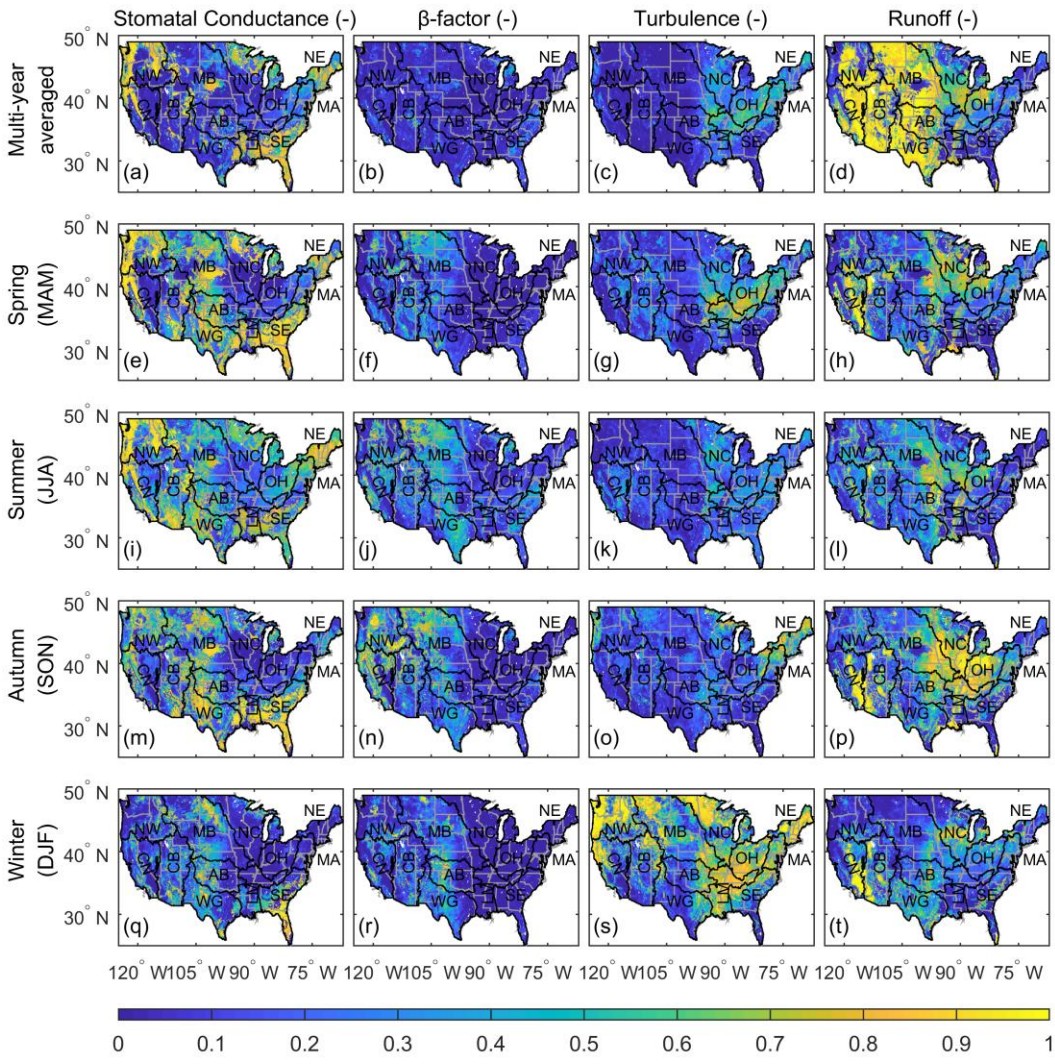

**Figure 14:** The Sobol' total sensitivity of the multi-year-averaged and seasonal (spring—MAM, summer—JJA, autumn—SON, winter—DJF) ET to the four parameterizations: stomatal conductance, soil moisture limitation to transpiration (β-factor), turbulence, and runoff. Higher values indicate higher sensitivities.

1085