# Peer review of "An ensemble of 48 physically perturbed model estimates of the $1/8^{\circ}$ terrestrial water budget over the conterminous United States, 1980-2015"

_Earth System Science Data, 2022_

## Author Response (AR1)

**Reply to RC1:**

The manuscript presents the 48-member Noah-MP simulations in CONUS and the evaluation results. Common terrestrial water budget variables are provided. A comprehensive evaluation is performed based on multi-source reference dataset. The manuscript is well written. The dataset will be useful for diverse applications. I think the manuscript is suitable for publication on ESSD. Below are some comments which could be useful to the authors.

RC: The dataset is not developed by this paper according to the description in the manuscript. For example, Fei et al. (2021), which is a publication of the same authors, already evaluated part of the 48-member Noah-MP model outputs. However, the description in the manuscript is kind of misleading (e.g., the abstract and line 79), making the readers have an impression that this manuscript runs the ensemble simulation. I recommend that the authors re-organize relevant contents, clearly stating the development and evaluation history of the 48-member simulations, and the role of this manuscript (e.g., evaluation and data release?) in the introduction part.

AR: Thanks for the suggestion. We have revised the abstract, introduction, and conclusions for clarification: (1) in the abstract, we revised the text as "this paper describes a dataset simulated from an ensemble of 48 physics configurations of Noah-MP" rather than "this study presents a 48-member ensemble simulation." (2) in the introduction, L79 is revised as "we have enriched the NLDAS-2 four-model ensemble ...", (3) L91 is deleted, (4) we added a sentence at the beginning of the last paragraph of the introduction "We have previously evaluated runoff and compared it with NLDAS. This paper describes the estimation of all the TWB variables." and (5) in the conclusions, the first sentence is revised as "this paper describes a dataset of TWB over the CONUS."

RC: Line 104: Can you talk more about the "pitfalls"?

AR: For instance, Fei et al. (2021) found that the ensemble members generated by naive perturbation of the Noah-MP physics are not independent enough from each other. The low independence hinders the skill gained from the ensembling method. The finding suggests that advanced techniques of physics perturbations should be developed to maximize the ensemble skill and minimize the ensemble size. We added these sentences in the revision.

RC: Section 2.2: Why these parameterizations are chosen? Can they represent the full range of uncertainty? Besides, I think the introduction to parameterizations can be moved the appendix. As a dataset description paper, these technical details could weaken the readability of the paper for most readers.

AR: The processes are selected as they directly control runoff generation and evapotranspiration and have shown their importance in global simulations. Limited by computational resources, we have not perturbed the parameterization of the cryosphere hydrological processes such as snow albedo (Chen et al., 2014,

doi:10.1002/2014JD022167; He et al., 2019, doi:10.1029/2019JD030823) and rain-snow partitioning (Wang et al., 2019, doi:10.1029/2019gl085722), which may limit the usage of the dataset in cryosphere hydrology. Section 2.2 is revised to clarify the criteria. The results show that the uncertainty range would be sufficient for variables other than transpiration, groundwater storage, and snow water equivalent. The discussion is added in Section 4.1 and the conclusions. Technical details are moved to Appendix A.

RC: Line 279: Is this recursive spin-up in a single year?
AR: Yes. We revised the sentence for clarification.

RC: I have some doubts about Sections 3.1 and 3.2. I think there is a mistake. In Eq 34, you should not subtract r_clim (see Eq 8 in Dirmeyer et al., 2006). Otherwise, r_clim is subtracted twice in Eq 37. For the subscript t in Section 3.2, I did not find any explanation (please correct me if I made a mistake). To be honest, the two sections use more equations and symbols than Dirmeyer et al., (2006) but make the same concept much less straightforward and harder to understand. Probably the authors want to use more symbols to make the definition clearer, but it turned out making things worse from my opinion. I suggest that the authors reorganize these sections.
AR: Thanks for pointing out the error. We revised Sections 3.1 and 3.2. r_clim is deleted from the two equations. It is a typing error and does not affect the analyses. We revised the two sections and used fewer symbols for clarification.

RC: Section 3.5: I understand that the reference datasets are important. But this section is too long for a dataset description dataset on ESSD. This can be a distraction from your core dataset. I am wondering whether you can remove some contents or move some contents to the appendix.
AR: Thanks for the suggestion. We condensed the contents in the revision and moved the correction of TWS in Appendix B.

RC: Line 492: Can you explain it more clearly?
AR: We corrected the statement here. In NE, MA, and OH, Noah-MP underperforms NLDAS in both the annual cycle and interannual anomaly. The underperformance is mainly due to Noah-MP having a higher variability than GRACE. The Differences in the variability between Noah-MP and GRACE could be resulted from: (1) Noah-MP overestimated the variability due to unsuitable parameter values. For instance, specific yield is an important parameter. The parameter is calibrated to 0.2 from a global simulation. The parameter value may not be suitable for these RFCs. (2) GRACE may underestimate the variability in these coastal RFCs. The data experience signal leakage from the ocean. The leaked signal can lower the temporal variability. We revised the abstract and conclusions accordingly.

RC: Figures 5 and 6: According to the second column, Noah-MP EM is not notably better than NLDAS EM. Can you explain how this affects the results in the third column? Besides, the statement "four estimates' arithmetic average outperforms the three-model

NLDAS ensemble mean at almost every NASMD site" is not always true (e.g., Fig. 6c and 6f). I suggest adding some quantitative statistics in the figures (e.g., the median value, or the ratio of positive values). This will make the comparison more straightforward.

AR: We added the ratio of positive value in the second and third columns. "Almost every" is changed to "most." The performance of Noah-MP EM significantly affects the results of the third column. As added in the revision, if the Noah-MP ensemble mean already outperforms the NLDAS ensemble mean, the ratio of the positive values in the third column is approximately 100%. If the Noah-MP ensemble mean underperforms the NLDAS ensemble mean, the ratio is approximately one-third (one-fourth) for the annual cycle (interannual anomaly).

RC: Figure 7: The figure caption is unclear. Besides, I think you mean "difference" (Line 524) instead of "relative bias" in the figure caption.

AR: Yes, it is "difference". Thanks. The caption is revised.

RC: Line 530-532: Any explanation?

AR: We added a Quantile-Quantile plot in the revised Figure 7. The plot shows that both Noah-MP and the NLDAS models tend to underestimate SWE in most areas but overestimate it when snow is extremely thick (SWE > 400 mm). Noah-MP performs better than the NLDAS models in most cases. The superiority of Noah-MP is likely attributable to the tree-layer snowpack module, which can provide a more smooth transition from shallow to thick snow than the single-layer Noah and Mosaic snow module and the quasi-two-layer VIC snow component. Detailed examination of the spatial correlation reveals that the superiority of Noah-MP over the NLDAS models appears in all elevation bands but is the most significant between 1000-2000 m with a spatial correlation of 0.85 versus 0.38. If the elevation is below 1000 m (above 2000 m), the spatial correlations are 0.77 versus 0.76 (0.89 versus 0.75). The discussion is added in the revision.

**Reply to RC2:**

The authors generated a 48-member perturbed-physics ensemble dataset configured from the widely-used Noah-MP LSM with a 0.125deg spatial resolution over CONUS. This new dataset includes major terrestrial water storage component terms as output, which can serve as an augmentation of the existing NLDAS-2 model ensemble. The authors also presented an evaluation of their model ensemble results. Overall, the manuscript is well written. Before it can be considered for publication, I have a few suggestions and comments for the authors to consider.

Specific comments:

RC: Which Noah-MP model code version was used in this study? Noah-MP has gone through a lot of updates in the past few years and is currently in version 4.4. Does the Noah-MP model version used in this study include those recent updates, such as a new roughness sublayer canopy turbulence scheme (Abolafia-Rosenzweig et al., 2021: https://doi.org/10.1029/2021MS002665), a new plant hydraulics scheme (Li et al., 2021: https://doi.org/10.1029/2020MS002214), and new snowpack parameter enhancements (He et al., 2021: https://doi.org/10.1029/2021JD035284). If not, I would suggest the authors at least briefly discuss these relevant Noah-MP updates and clarify the model code version they used in this study.

AR: We used Noah-MP along with WRF 3.6. It is basically the same as the model described by Niu et al., 2011, JGRA with bug fixes. We did not include the above-mentioned updates. We made the limitations explicit in the revised Section 2.2 and discussed possible improvements in the Conclusion.

RC: Line 25: Please also provide the temporal resolution for the dataset (e.g., hourly output?).

AR: The temporal resolution is monthly. We added it in the revision.

RC: Line 110: "Section 0 concludes this study." What is Section 0? A typo?

AR: It is a typo. Should be Section 6. Corrected in this revision.

RC: Is there a way to quantitatively present the ensemble uncertainty range in the abstract and conclusion sections (e.g., x% of mean)? This will be very informative.

AR: We added a column in Table 2, showing the percentage of the ensemble spread relative to the climatological mean. The ensemble spread is largest for the surface runoff (34%) and smallest for the snow water equivalent (2.5%). Section 4.1 is revised to reflect the changes. A paragraph is added in the Conclusion Section, briefing the uncertainty range.

RC: What criteria were used to select and include only the four Noah-MP processes (i.e., runoff, stomatal conductance, soil moisture factor, turbulence) in the ensemble? What about other related processes such as snowpack-related schemes? For example,

previous studies (e.g., Chen et al., 2014: https://doi.org/10.1002/2014JD022167; He et al., 2019: https://doi.org/10.1029/2019JD030823) have suggested that snow albedo schemes could play an important role in affecting Noah-MP snowpack water processes. AR: The processes are selected as they directly control runoff generation and evapotranspiration and have shown their importance in global simulations. Limited by computational resources, we have not perturbed the parameterization of the cryosphere hydrological processes such as snow albedo (Chen et al., 2014, doi:10.1002/2014JD022167; He et al., 2019, doi:10.1029/2019JD030823) and rain-snow partitioning (Wang et al., 2019, doi:10.1029/2019gl085722), which may limit the usage of the dataset in cryosphere hydrology. We revised Section 2.1 to clarify the criteria and added a paragraph at the end of the Conclusions to clarify the limitation.

RC: Line 160: Where did these parameter values come from? Based on previous observations? Similarly, some clarifications are needed for the parameter values used for other schemes in this study.
AR: The parameters use the Noah-MP default values. We revised the manuscript to clarify this.

RC: Lines 224-225: I believe the q_sat and q_a are mixing ratios instead of specific humidity. There are some typos in the Noah-MP code comments regarding this. Please double check.
AR: Thanks for the comment. We have checked both the code and the literature. The code differs from the literature. In Jacquemin and Noilhan (1990, https://doi.org/10.1007/BF00123180) (which Chen et al. (1996) referred back), they are specific humidity. Whereas in Noah-MP, they are mistakenly implemented as mixing ratio. We have revised the sentence to illustrate the difference.

RC: Line 280: Although details can be referred back to these two previous studies, I would suggest providing a brief description of the necessary information here, e.g., how many years of spin-up.
AR: We revised the sentence for clarification: "The initial states on 1 January 1980 were obtained by cycling the year 1979 one hundred times."

RC: Line 397: Bilinear interpolation from 1-km to 0.125deg may not be a good idea. A better way is to aggregate/average all 1-km pixels within each 0.125deg pixel.
AR: Thanks. In the revision, we aggregated the 1-km SNODAS to each 0.125deg NLDAS grid and re-analyzed the results.

RC: Section 4.1: I would expect more detailed quantitative descriptions and discussions regarding the ensemble spread and differences and causes for their differences, because this large model ensemble is the key of this new dataset. The current description is too brief and qualitative.
AR: We revised the Results section to clarify the organization of the descriptions. Section 4.1 aims to compare the ensemble spread among different variables and

between the annual cycle and interannual anomaly. Sections 4.2 to 4.5 present both the skill and ensemble spread. The ensemble spread relative to the climatological mean and temporal variability is calculated and presented in Table 2. Discussions of the ensemble spread is added in Section 4.1.

RC: As the authors mentioned, the Noah-MP ensemble spread is relatively small for soil moisture (Fig.4) and SWE (Fig. 8), would this be caused by too similar physical formulations of the Noah-MP schemes or parameters tuned by previous studies or something else?

AR: We discussed the small spread in soil moisture and SWE in the revised Sections 4.1, 4.3, and 4.4. Whether a spread is adequate depends on the bias. For SWE, the ensemble spread seems adequate in RFCs such as NE (Figure 8) since the bias is small. On the other hand, the spread is too small in RFCs such as NW, CN, and CB. The small ensemble spread in SWE is likely due to inadequate sampling of the feasible physics parameterizations such as snow albedo, rain-snow partitioning, subgrid heterogeneity, and roughness length. For soil moisture, the Noah-MP ensemble spread appears too small in AB and CB relative to the bias (Figure 4). Mosaic has a detailed consideration of subgrid variability and outperforms Noah-MP. The Noah-MP ensemble spread could be enlarged to consider the uncertainty associated with spatial heterogeneity. On the whole, it is likely the NLDAS models have too much spread, considering that VIC simulated soil moisture using a conceptual water tank.

RC: I would suggest adding uncertainty bars for observational points in the evaluation figures (e.g., Figs. 1, 4, 8, 10).

AR: We added error bars in the figures. The error bars in Figures 1, 4, 8, and 10 are calculated as the standard deviation of the year-to-year differences.

---

## Author Response (AR2)

We are deeply thankful to the editor and the editorial support team for your work on the manuscript.

As you suggested, we increased the size of the figures (Figures 1, 2, 3, 4, 5, 6, 8, 9, 10, 11, 13, and 14).

Besides, we made minor adjustments to the manuscript.

---

## Author Response (AR3)

Response to Reviewers

We are deeply thankful to the editors for the constructive comments that greatly improve the quality of this manuscript. We have addressed the comments one by one as follows.

*RC*: Even in the new version, I think that Figures 5, 6, 9, 11, 13, 14 are too small. Furthermore, including the RFC-codes in every single map creates a lot of additional "clutter" that makes the Figures very hard to read. So I would either increase the size even more so that we can easily identify the RFC-codes when your manuscript is printed on normal A4-paper or remove the codes from these Figures and add an additional map with only the boundaries and codes of the River Forecast Center domains. But please do not increase the size of the color scales in Figures 13 and 14 as they already look too long.

*AR*: We have increased the size of Figures 5, 6, 9, 11, 13, and 14. They are now 17 cm wide. The size can fit into an A4 paper with 2 cm margins. We also deleted the RFC codes from almost every but the one last panel of these figures. We hope the revised figures are easier to read. Besides, placing the RFC codes inside the figures should be convenient for reference from the text.

*RC*: You seem to use the same bounding box and a simple "lat-lon projection" in every single map. In such cases, please use a lat-lon-ratio of 1:1 (i.e., your grid cells / pixels are always squares) for every Figure as otherwise, your maps look quite inconsistent.

*AR*: We have adjusted the aspect ratio of every map to 1:1.

*RC*: I would also either remove the title "Total time series" from Figure 4 or add something more meaningful like "Monthly averages" or "Monthly time series".

*AR*: We have revised the title "total time series" to "monthly time series."

We also want to clarify the changes in authorship here. The changes were made in the first revision (Submitted on 19 Feb 2023). An author, Shu Wang, was added to the author list. She analyzed the snow-related variables and rewrote the Section 4.4 according to the reviewers' comments. The authorship has been modified to reflect the changes.